# Click beetle luciferase mutant and near infrared naphthyl-luciferins for improved bioluminescence imaging

Mary P. Hall[1], Carolyn C. Woodroofe[1,6], Monika G. Wood[1], Ivo Que[2], Moniek van 't Root[3], Yanto Ridwan[3,4], Ce Shi[5], Thomas A. Kirkland[5], Lance P. Encell[1], Keith V. Wood[1], Clemens Löwik[3,4] & Laura Mezzanotte ®[3,4]

The sensitivity of bioluminescence imaging in animals is primarily dependent on the amount of photons emitted by the luciferase enzyme at wavelengths greater than 620 nm where tissue penetration is high. This area of work has been dominated by firefly luciferase and its substrate, D-luciferin, due to the system's peak emission (~ 600 nm), high signal to noise ratio, and generally favorable biodistribution of D-luciferin in mice. Here we report on the development of a codon optimized mutant of click beetle red luciferase that produces substantially more light output than firefly luciferase when the two enzymes are compared in transplanted cells within the skin of black fur mice or in deep brain. The mutant enzyme utilizes two new naphthyl-luciferin substrates to produce near infrared emission (730 nm and 743 nm). The stable luminescence signal and near infrared emission enable unprecedented sensitivity and accuracy for performing deep tissue multispectral tomography in mice.

[1] Promega Corporation, Madison, WI 53711, USA. [2] Department of Radiology, Leiden University Medical Center, 2300 RC Leiden, The Netherlands. [3] Optical molecular imaging, Department of Radiology & Nuclear Medicine, Erasmus University Medical Center, 3000 CA Rotterdam, The Netherlands. [4] Department of Molecular Genetics, Erasmus University Medical Center, 3000 CA Rotterdam, The Netherlands. [5] Promega Biosciences Incorporated, San Luis Obispo, CA 93401, USA. [6] Imaging Probe Development Center, National Heart, Lung, and Blood Institute, National Institutes of Health, Rockville, MD 20850, USA. Mary P. Hall and Carolyn C. Woodroofe contributed equally to this work. Correspondence and requests for materials should be addressed to L.M. (email: l.mezzanotte@erasmusmc.nl)

Bioluminescence imaging (BLI) using firefly luciferase (Luc2) and D-luciferin (D-LH2) has become a standard method for gene expression analysis and preclinical evaluation of potential therapies in mouse models[1, 2]. The Luc2/D-LH2 system has been broadly adopted because the light it produces peaks near 600 nm at 37 °C and can adequately penetrate shallow tissues such as skin. However, in deeper tissues such as lung, brain, and bone, the sensitivity of Luc2/D-LH2 is limited due to absorption by hemoglobin, melanin, and other tissue components[3, 4]. In addition, the biodistribution of D-LH2 is often insufficient for sustained imaging in challenging tissues, such as brain[5].

To improve resolution for deep tissue imaging, attempts have been made to shift the wavelength of bioluminescence emission into the near infrared (NIR) (650–900 nm). Mutagenesis has been used successfully to red-shift the spectral properties of luciferases (utilizing D-LH2 as substrate), but mutants with a significant NIR component to their emission have been elusive[6, 7]. This is likely

an inherent limitation of the actual photon-emitting species, oxyluciferin[6, 8–10].

Analogs of D-LH2 with extended π conjugation to support longer wavelength photon generation have been developed that produce NIR bioluminescence with Luc2[11–14], and aminoluciferin-NIR dye conjugates have been shown to produce NIR signals via energy transfer[15]. Kuchimaru et al. recently described a new substrate, AkaLumine-HCl (Aka-HCl), that contains extended conjugation and produces NIR bioluminescence (677 nm peak emission)[16]. However, the utility of these substrates is still limited due to the fact that bioluminescence signals are only enhanced over Luc2/D-LH2 at limited substrate concentrations.

We addressed the challenges associated with deep tissue imaging by creating improved substrates and luciferases. We designed two naphthyl-based luciferin analogs, amino-naphthyl naphtho[2,1]thiazole luciferin (NH2-NpLH2) and hydroxy-

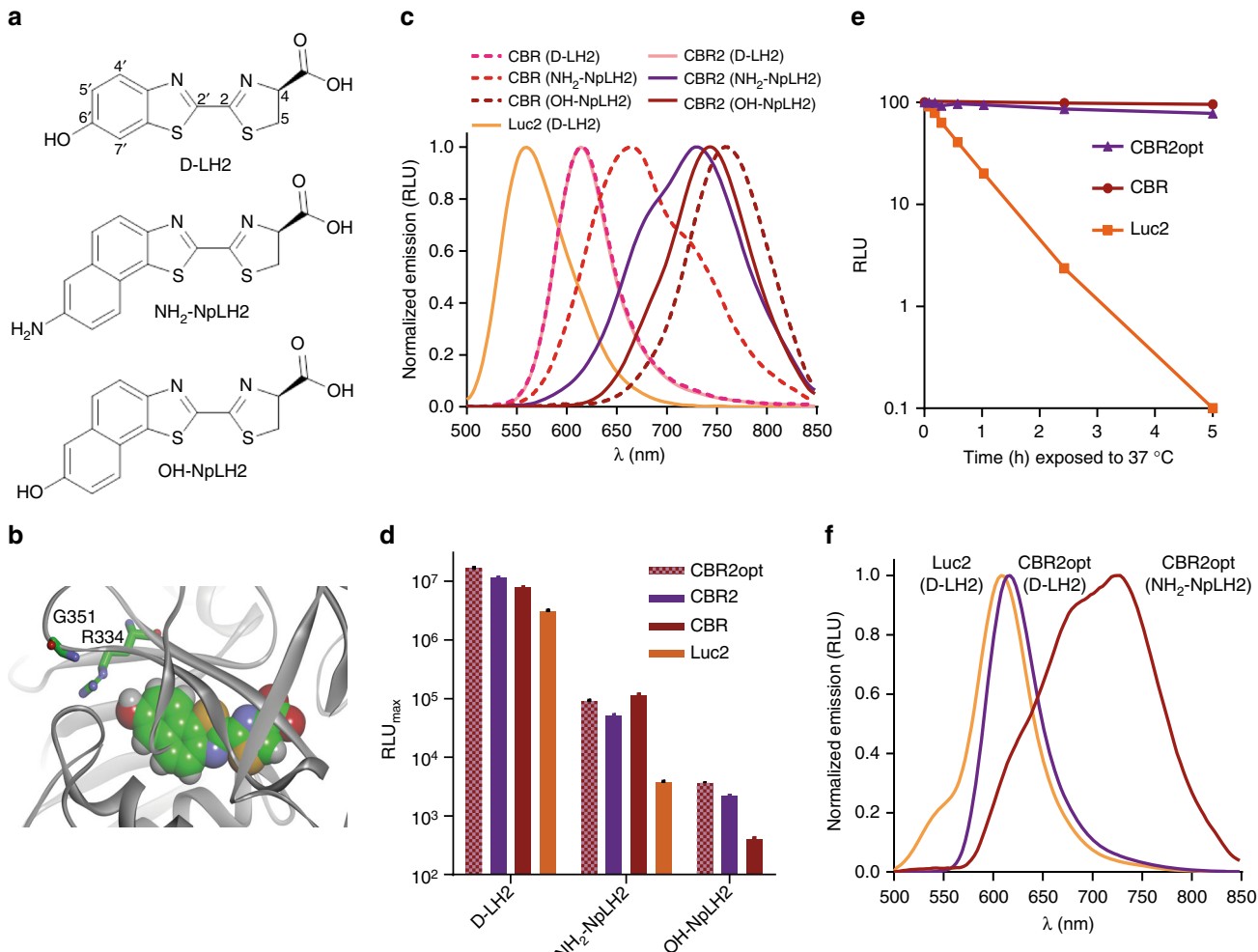

**Fig. 1** Biochemical and preliminary cell-based characterization of naphthyl-luciferin bioluminescence substrates and different luciferases. **a** Chemical structures of NH2-NpLH2, OH-NpLH2, and D-LH2. The naphthyl moieties (i.e., naphtho[2,1]thiazole) provide additional π conjugation. **b** Homology model of CBR highlighting residues R334 and G351. **c** Bioluminescence emission spectra of D-LH2, NH2-NpLH2, and OH-NpLH2 produced by purified CBR and CBR2 (data presented as means ($n = 3$) ± S.D.). Peak values for CBR are reported in Table 1. For CBR2 the peak values are as follows: D-LH2, 614 nm; NH2-NpLH2, 730 nm; OH-NpLH2, 743 nm. The spectrum for Luc2/D-LH2 (peak emission 559 nm) is shown for reference. **d** Live cell (HEK-293) bioluminescence intensity ($RLU_{max}$) for different combinations of substrate and luciferase ($n = 3$). CBR2opt is a gene encoding the same CBR2 enzyme as the CBR2 gene, but it uses codons optimized for expression in mammalian cells. There was no detectable signal for Luc2/OH-NpLH2. **e** Physical stability of CBR, CBR2 (encoded by CBR2opt) and Luc2 in HEK-293 lysates at 37 °C ($n = 3$). **f** Bioluminescence emission spectra of D-LH2 and NH2-NpLH2 produced by CBR2opt and Luc2 cells (HEK-293). Emission peaks: Luc2/D-LH2, 608 nm; CBR2opt/D-LH2, 617 nm; CBR2opt/NH2-NpLH2, 728 nm. The signal for Luc2/NH2-NpLH2 was too low to generate meaningful spectral data. Attempts to collect spectra for OH-NpLH2 were also unsuccessful due to insufficient signal. Error bars represent standard deviation (S.D.)

**Table 1 Substrate characteristics**

| | Emission peak[a] (nm) | | | Brightness[b] (RLU) | | |
|---|---|---|---|---|---|---|
| | D-LH2 | NH$_2$-NpLH2 | OH-NpLH2 | D-LH2 | NH$_2$-NpLH2 | OH-NpLH2 |
| Luc2 | 559 | 678[c], 719[d] | * | 1.0 | $1 \times 10^{-5}$ | (−) |
| UltraGlo | 556 | 555[c], 667[d] | 659 | 0.5 | $1 \times 10^{-4}$ | $4 \times 10^{-6}$ |
| CBG99 | 546 | 544 | * | 1.0 | $2 \times 10^{-2}$ | $2 \times 10^{-6}$ |
| CBR | 614 | 664 | 758 | 0.2 | $2 \times 10^{-4}$ | $3 \times 10^{-6}$ |

* Could not be obtained due to low signal. (−) undetectable signal
[a] Average peak values (±3 nm) as determined from multiple (≥3) spectral reads
[b] Calculated brightness data normalized to Luc2/D-LH2 (1.0); Reactions consisted of equal volumes of 1 µg mL$^{-1}$ enzyme (in TBS (pH 7.5) containing 0.01% BSA) and 150 µM substrate+1 mM ATP (in 150 mM HEPES (pH 7.5), 1 mM CDTA, 16 mM MgSO$_4$, and 1% NP-9); n = 3. Signal values for NH$_2$-NpLH2 and OH-NpLH2 were lower than D-LH2 partly because of luminometer PMT bias for shorter wavelength photons
[c] major peak
[d] minor peak

naphtha[2,1]thiazole luciferin (OH-NpLH2), and evaluated these substrates using several beetle luciferase enzymes to find the most compatible pairing. Both substrates produced NIR bioluminescence with click beetle red luciferase (CBR)[17], but signals were weak compared to Luc2/D-LH2. To improve luminescence intensity we used rational enzyme design and codon optimization to engineer a mutant luciferase, CBR2, encoded by a codon-optimized gene sequence, CBR2opt. In cells the mutant produced significantly more signal with the OH-NpLH2 substrate compared to CBR. Although light output with NH$_2$-NpLH2 (the brighter of the two analogs) was essentially unchanged, the emission spectrum shifted dramatically (~ 65 nm) into the NIR (730 nm peak). In addition to providing improved signal for OH-NpLH2 and a red-shift for NH$_2$-NpLH2, the CBR2 enzyme was also more stable in live cells compared to Luc2. This suggested that it could provide greater light output by accumulating to higher levels when expressed in animals.

Herein, we demonstrate that the mutant click beetle luciferase and NH$_2$-NpLH2 each represent significant advancements for in vivo BLI. The mutant maintains the ability to utilize D-LH2 as a substrate, and this pairing provides improved sensitivity in mice compared to Luc2/D-LH2. Further, when testing for deep tissue multispectral tomography, the pairing of the mutant enzyme with NH$_2$-NpLH2 produces highly resolved NIR signals which enable a precise 3D diffuse tomographic reconstruction for localization of cells in the brain using NIR emission filters.

## Results

**Characterization of NIR naphthyl-luciferins.** It was previously demonstrated that extension of π conjugation in luciferin analogs reduces the HOMO-LUMO energy gap in corresponding oxyluciferins to result in a red-shifted spectrum[11–14]. We envisioned that the fusion of an additional phenyl ring to the benzothiazole fragment of LH2 could increase conjugation and create a substrate capable of producing longer wavelength light. Compounds with extended conjugation arising from unsubstituted polyolefins (e.g., cyanine dyes) are often prone to chemical and photoinstability[18]. To extend the conjugation of our luciferin we turned to naphthothiazole-based analogs rather than luciferin analogs that have additional alkene units between the aryl and thiazoline moieties (e.g., Aka-HCl). We identified NH$_2$-NpLH2 and OH-NpLH2 (Fig. 1a) as candidates for the production of NIR bioluminescence with an optimized luciferase. These analogs were conveniently synthesized using conventional luciferin chemistry with high enantiopurity (see Supplementary Figs. 1–3, Supplementary Table 1). The monopotassium salt forms of NH$_2$-NpLH2 and OH-NpLH2 were also formulated and found to have good aqueous solubility (NH$_2$-NpLH2, 46 mM; OH-NpLH2, 50 mM) and stability at ambient temperature in PBS for at least 24 h. In

addition, neither compound was cytotoxic (HEK-293, Hela) or showed any acute toxicity or adverse effects in mice when administered intraperitoneally at high concentration[19].

NH$_2$-NpLH2 and OH-NpLH2 were tested as substrates with Luc2, click beetle green (CBG99)[17], CBR, and UltraGlo[20]. Each enzyme displayed a range of spectral properties and bioluminescence intensities (Table 1). NH$_2$-NpLH2 was utilized by all of the enzymes tested, and Luc2, CBR, and UltraGlo each produced spectral emission peaks in the NIR (~ 655–720 nm). In contrast, OH-NpLH2 was only utilized by CBR and UltraGlo. Spectral data were elucidated for OH-NpLH2, but only under conditions of high enzyme concentration. CBR and UltraGlo both gave significant emission in the NIR with this substrate. However, the peak for OH-NpLH2 with CBR is particularly striking (758 nm), as it is 52 nm longer than what was previously reported as the most red-shifted bioluminescence system to date[13]. Although both naphthyl-luciferin substrates produced NIR emission, signal intensities were generally much lower (~5000–500,000-fold) than Luc2/D-LH2 (note reduced spectral sensitivity for longer wavelength photons contributed to the lower relative signals for the NIR analogs[21] as measured using a PMT-based luminometer[22]). The exception was CBG99/NH$_2$-NpLH2. This pair produced the highest signal using a luminometer, but had the shortest peak emission wavelength (544 nm) of all combinations examined. CBR presented the best opportunity for achieving both high luminescence intensity and NIR emission and was therefore selected for optimization with the naphthyl-luciferin substrates as a means to develop an improved system for animal imaging.

**Design of CBR2 luciferase and CBR2opt gene sequence.** To assist in the optimization of CBR for enhanced luminescence with the NIR substrates, we built a homology model of the enzyme (Fig. 1b). The model was based on the X-ray structure of a related firefly luciferase with bound 5′-O-[N-(dehydroluciferyl)-sulfamoyl] adenosine (DLSA)[23] (PDB code 2D1S), but we replaced the dehydroluciferin moiety of DLSA with OH-NpLH2 (chosen because of its significant red-shift). Efficient light emission for beetle luciferases is thought to require a hydrogen acceptor for the 6′-hydroxyl group of LH2 (Fig. 1a), and the source for this is likely the conserved arginine (R334 in CBR) located at the base of the luciferin binding pocket. R334, which also participates in a stabilizing hydrogen-bonding network that helps shield luciferin from solvent, has been implicated in modulating the level of light output from 6′-substituted aminoluciferins[24]. Together these observations indicated R334 was a suitable target for mutagenesis.

Based on our analysis of the CBR model, we targeted two nearby active site residues in combination, R334 and the directly opposing G351 (Fig. 1b). We hypothesized that replacement of these residues with different combinations of hydrogen acceptors

**Table 2 Kinetic parameters for beetle luciferases and substrates**

|  | Brightness[a] ($RLU_{max}$ $\mu M^{-1}$) | | $K_m$ ($\mu M$) | | | |
|---|---|---|---|---|---|---|
|  | D-LH2 | $NH_2$-NpLH2 | D-LH2 | $NH_2$-NpLH2 | ATP (saturating D-LH2) | ATP (saturating $NH_2$-NpLH2) |
| Luc2 | $3 \times 10^9$ | * | 1 | * | 286 | * |
| CBR | $6 \times 10^8$ | $6 \times 10^5$ | 10 | 0.3 | 159 | 132 |
| CBR2 | $4 \times 10^8$ | $8 \times 10^4$ | 22 | 0.5 | 305 | 112 |

* Value could not be determined because of insufficient signal above background at low substrate concentrations
[a] Brightness ($RLU_{max}$ $\mu M^{-1}$ enzyme) and $K_m$ calculated from substrate titrations using Michaelis–Menten regression analysis; $n = 3$. Brightness values for $NH_2$-NpLH2 were lower than D-LH2 partly because of luminometer PMT bias for shorter wavelength photons

and donors could improve enzyme efficiency by restructuring the hydrogen-bond network to better accommodate the new substrates. Furthermore, mutagenesis at these sites offered the potential to provide an alternate hydrogen-bond acceptor for OH-NpLH2. To test our hypothesis, we constructed a library of mutational combinations at positions 334 (E, Q, D, N, H, S, T, C, Y) and 351 (R, K, E). The library was expressed in bacteria and screened as lysates for improved light output using each NIR substrate. The mutant enzyme of highest interest, R334S+G351R (CBR2), was purified and further characterized. Subsequently, the gene sequence for CBR2 was optimized (CBR2opt; 79% identity to CBR2) to match codon frequencies as found in mice (see Supplementary Fig. 4).

**Characterization of CBR2 with D-LH2 and naphthyl-luciferins**. As a means to investigate their biochemical properties, CBR and CBR2 were purified. CBR2 produced luminescence comparable to CBR with OH-NpLH2 ($3 \times 10^{-6}$ RLU, as reported in Table 1), and the spectral emission peak was slightly blue-shifted to 743 nm (Fig. 1c). Surprisingly, the peak emission for CBR2/$NH_2$-NpLH2 was shifted by more than 65 nm (to 730 nm). CBR2/D-LH2 produced ~ 2-fold less signal compared to CBR/D-LH2, and peak emission (614 nm) was unchanged.

Table 2 summarizes the results of the biochemical kinetics studies. Briefly, the specific activities for CBR and CBR2 utilizing D-LH2 as substrate were 5- and 7.5-fold lower than that of Luc2, and their affinities for D-LH2 were 10- and 22-fold weaker. In contrast, CBR and CBR2 had 30- and 44-fold higher affinity for $NH_2$-NpLH2 compared to D-LH2. Although D-LH2 utilization differed significantly between enzymes, affinities for ATP were similar. While it was possible to obtain a measurable signal for Luc2/$NH_2$-NpLH2, Luc2/OH-NpLH2, CBR/OH-NpLH2, and CBR2/OH-NpLH2 using a large excess of enzyme, accurate kinetic parameters could not be determined due to rapid signal decay and low signal to background under these conditions.

Although in a biochemical setting Luc2/D-LH2 was brighter than both CBR and CBR2, this was not the case in live cells at 37 °C. CBR, CBR2, and CBR2opt (mammalian codon optimized version of CBR2; encodes the same CBR2 enzyme sequence) each produced an enzyme capable of generating 2–3-fold more luminescence with D-LH2 compared to Luc2 in HEK-293 cells (Fig. 1d). This was likely a result of higher accumulation of enzyme in cells contributed by greater physical stability at 37 °C for the CBR and CBR2 enzymes ($t_{1/2} > 10$ h) compared to Luc2 ($t_{1/2} < 30$ min) (Fig. 1e). The difference in brightness between Luc2/D-LH2 and CBR2/$NH_2$-NpLH2 with purified enzyme at ambient temperature was nearly 10,000-fold (Tables 1 and 2). However, when expressed in cells the difference in signal between CBR2opt/$NH_2$-NpLH2 and Luc2/D-LH2 narrowed to only 33-fold. For CBR/CBR2/CBR2opt this was likely due to improved expression and stability at elevated temperature. As observed previously (Table 1), there was no detectable signal for Luc2/OH-NpLH2. The calculated $K_m$ values for all three substrates were

higher in cells (Supplementary Fig. 5), but relative affinities were consistent with the biochemical data.

We also measured bioluminescence spectra in cells that expressed either CBR2opt or Luc2 (Fig. 1f). At 37 °C the signal for cells treated with OH-NpLH2 was too low to collect accurate spectra, as was the signal for Luc2 cells treated with $NH_2$-NpLH2. CBR2opt expressing cells produced emission peaks of 617 nm and 728 nm with D-LH2 and $NH_2$-NpLH2, respectively. This is in agreement with the data generated using purified enzyme (Fig. 1c). Note the emission peak for Luc2/D-LH2 under these conditions (i.e., in cells at 37 °C) was red-shifted to 608 nm, a value consistent with previous reports[25].

To evaluate the ability of $NH_2$-NpLH2 and D-LH2 to permeate cell membranes, we measured the bioluminescence of both substrates in either intact or lysed HEK-293 cells expressing either Luc2 or CBR2opt. The ratio of the signals produced from each condition was calculated as a means to assess relative permeability. The intact/lysed ratio was greater than 10-fold higher for $NH_2$-NpLH2 than for D-LH2, which suggests the new substrate permeates HEK-293 cell membranes more efficiently than D-LH2 (Supplementary Fig. 6). To determine if the method of lysis contributed to the results, we examined both digitonin and Passive Lysis Buffer (PLB). Similar results were obtained regardless of the lytic procedure used.

**Characterization in cells**. To verify the biochemical data and results from transient transfections, we created two stable expressing HEK-293 cell lines by lentiviral transduction: (1) HEK-EF1-Luc2-T2A-copGFP, and (2) HEK-EF1-CBR2opt-T2A-copGFP. The lentiviral constructs contained either Luc2 or CBR2opt gene sequences fused to copGFP (linked by a picornavirus T2A peptide sequence)[26]. Each cell line was sorted twice using the GFP signal, and average expression was quantified using in cell Western blotting and fluorescence microscopy (Fig. 2a, b).

We evaluated bioluminescence emission using four substrates: D-LH2, $NH_2$-NpLH2, OH-NpLH2, and the recently developed NIR substrate, Aka-HCl[16]. Images were acquired using a small animal imaging system equipped with a back-illuminated and cooled CCD camera[27, 28]. As shown in Fig. 2c, CBR2opt cells treated with D-LH2 produced the highest signal (similar to transient transfection data shown in Fig. 1d), but CBR2opt/$NH_2$-NpLH2 and Luc2/Aka-HCl cells also gave strong signals. It is important to note that the frequency of photons longer than 640 nm for CBR2opt/D-LH2 was higher than that of Luc2/D-LH2. However, all pairings tested except for Luc2/OH-NpLH2 (no signal could be detected) showed a high frequency of longer wavelength signal (Fig. 2d, e). Similar results were obtained using the alternative cell line, MCF-7, expressing the CBR2opt or Luc2 genes (Supplementary Fig. 7).

In summary, CBR2opt/D-LH2 gave signals approximately four times higher than Luc2/D-LH2 as extrapolated by the slope values of linear regression analysis performed on serial dilutions of transduced HEK-293 cells (Fig. 2f). On the basis of these results

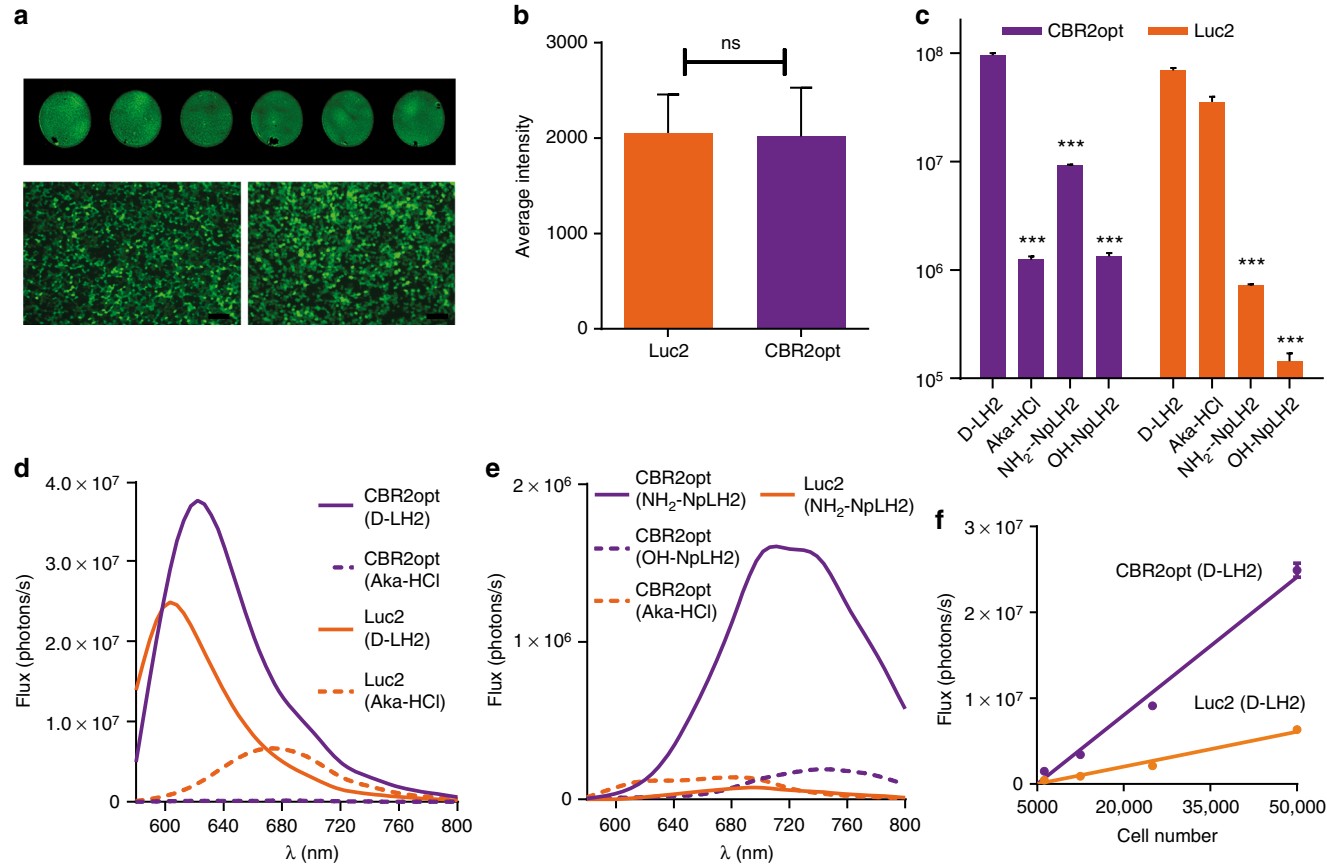

**Fig. 2** Characterization of naphthyl-luciferin substrates in stable luciferase cell lines. **a** In cell Western analysis (top) and fluorescence microscopy (bottom) of HEK-293 cells expressing Luc2 or CBR2opt (GFP fusions). Scale bar = 10 μm. **b** Quantification of Luc2/CBR2opt expression based on in cell (50,000 HEK-293 cells) Western analysis (data presented as means ($n = 3$) ± S.D. (ns not significant). **c** Bioluminescence emission (photon flux; CCD camera) for D-LH2, NH$_2$-NpLH2, OH-NpLH2, and Aka-HCl produced by Luc2 or CBR2opt expressed in HEK-293 cells; 10 min time point ($n = 3$). Each column is compared to CBR2opt/D-LH2 or Luc2-D-LH2 (***$p < 0.001$; ONE-Way Anova followed by Tukey's T test). **d** Live cell bioluminescence emission spectra of D-LH2 and Aka-HCl produced by CBR2opt and Luc2. **e** Live cell bioluminescence emission spectra of NH$_2$-NpLH2 and OH-NpLH2 produced by CBR2opt and Luc2. There was insufficient signal to collect spectra for Luc2/OH-NpLH2. CBR2opt/Aka-HCl is duplicated from **d**. **f** Bioluminescence signals for CBR2opt and Luc2 as a function of cell number ($n = 3$). Linear regression (for slope determination) was performed on data from samples producing signals above background. Error bars represent S.D.

we compared CBR2opt with D-LH2 or NH$_2$-NpLH2 to Luc2 with either D-LH2 or Aka-HCl for imaging applications in mice.

**Imaging skin of C57BL/6 black fur mice**. C57BL/6 black fur mice pose a challenge for imaging, even at shallow tissue depth, because of high absorption of photons by the dark fur. To investigate whether greater stability (and presumably expression/accumulation in cells) and longer wavelength emission with both D-LH2 and NH$_2$-NpLH2 would translate to improved imaging in black fur mice, we implanted $1 \times 10^6$ HEK-293 cells stably expressing either CBR2opt or Luc2 into the backs of animals and treated subcutaneously (150 mg kg$^{-1}$ for both D-LH2 and NH$_2$-NpLH2). The combination of CBR2opt/D-LH2 produced nearly 8-fold higher signal compared to Luc2/D-LH2 (Fig. 3a, b). It is likely that the superior signal for CBR2opt/D-LH2 compared to Luc2/D-LH2 under these presumably substrate saturating conditions is the combined result of a more stable enzyme and a greater proportion of light being generated > 650 nm.

It is worth noting that the signal from both enzymes when paired with NH$_2$-NpLH2 was only ~ 2-fold lower than Luc2/D-LH2. This difference is much less than that observed using enzymes or cells (Figs. 1 and 2), which suggests that the NIR photons produced by NH$_2$-NpLH2 are, as expected, less prone to

absorption by the black fur compared to signals coming from D-LH2.

**Deep brain imaging**. Imaging transplanted cells deep inside the brain using standard Luc2/D-LH2 presents a challenge not only because of the distance to the surface, but also because of its dark color, high cellular and molecular density, and relatively inefficient biodistribution of D-LH2. The bioluminescence produced by Luc2/D-LH2 generally contains an insufficient amount of longer wavelength photons to efficiently penetrate brain tissue and escape animals without being absorbed. To determine if the longer wavelength properties associated with CBR2opt/D-LH2 and CBR2opt/NH$_2$-NpLH2 could help overcome this limitation, we implanted $1 \times 10^5$ CBR2opt or Luc2 cells (stable HEK-293 lines) at 3 mm depth in mouse brains and imaged after intravenous administration of 300 mg kg$^{-1}$ of D-LH2 or 220 mg kg$^{-1}$ of NH$_2$-NpLH2 (Fig. 4a, b). The higher concentration for D-LH2 was chosen because, although less common than the standard 150 mg kg$^{-1}$ dose, it was previously shown to produce higher signals in vivo[29]. Intravenous administration was chosen instead of intraperitoneal administration to ensure higher availability of substrate in the brain[5]. The dose of NH$_2$-NpLH2 was chosen because it is close to the solubility limit in PBS. The bioluminescence signal from CBR2opt/D-LH2 was 2-fold higher than the

**a**

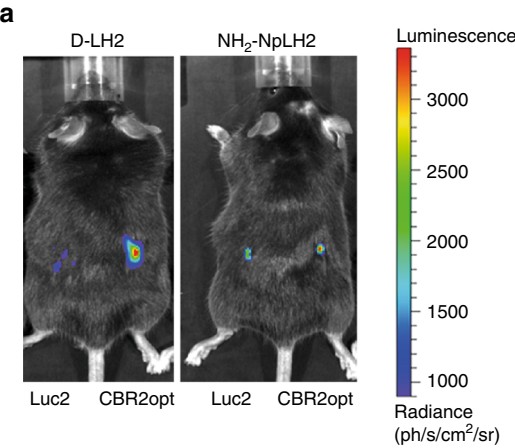

**b**

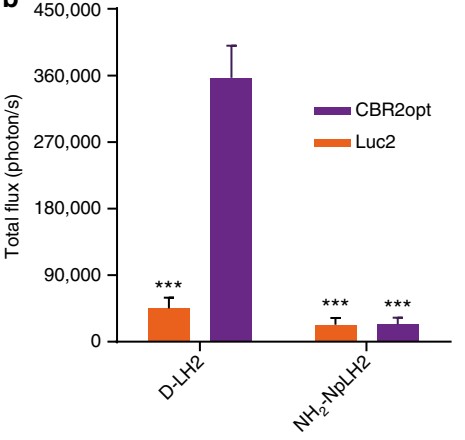

**Fig. 3** Imaging the backs of C57BL/6 black fur mice. **a** Representative bioluminescence images of mice after subcutaneous implantation of $1 \times 10^6$ of HEK-EF1-Luc2-T2A-copGFP and HEK-EF1-CBR2opt-T2A-copGFP cells and intraperitoneal injection (150 mg kg$^{-1}$) of either D-LH2 or NH$_2$-NpLH2 (data presented as means ($n = 3$) ± S.D.). **b** Quantification of flux (photon/s) with an exposure time of 60 s (***$p < 0.001$ compared to CBR2opt/D-LH2). Error bars represent S.D.

signal from Luc2/D-LH2 ($p$ value < 0.05; ONE-way ANOVA followed by a Tukey's t test; same test was performed throughout the text if not specified), and approximately 3-fold higher than the signal from CBR2opt/NH$_2$-NpLH2 ($p$ value < 0.01). These findings are consistent with our results showing the effectiveness of CBR2opt in the subcutaneous model (Fig. 3).

Note that we carried out similar experiments where substrate was administered intraperitoneally (Supplementary Fig. 8), but the detection kinetics were much slower compared to the intravenous dosing and we did not observe a significant difference between Luc2/D-LH2 and CBR2opt/NH$_2$-NpLH2.

Using the brain model, we also compared CBR2opt/D-LH2 to CBR2opt and Luc2 with the NIR substrate, Aka-HCl (administered at its maximum solubility in PBS, 50 mg kg$^{-1}$). CBR2opt/D-LH2 and Luc2/Aka-HCl produced essentially the same amount of signal (Fig. 4c, d). CBR2opt/Aka-HCl generated 6-fold lower signal intensity compared to CBR2opt/D-LH2 and Luc2/Aka-HCl ($p$ value < 0.01), indicating that Aka-HCl is a relatively poor substrate for CBR2opt.

**Bioluminescence tomography in mouse brain**. We speculated that the bright, stable, NIR emission of CBR2opt/NH$_2$-NpLH2 could improve the accuracy of bioluminescence tomography (BLT) compared to Luc2/D-LH2 (~ 600 nm). We compared

CBR2opt/NH$_2$-NpLH2 to the brightest D-luciferin based system to date, CBR2/D-LH2, for bioluminescence tomography using the brain model. The experiments were performed with intraperitoneal injection of substrates as a means to generate a stable light emission. The sustained signal allowed for the collection of photons over time using a series of band pass filters. The resulting images were then used to reconstruct the light source using an algorithm developed by Living Image 4.3 software. We co-registered CT images of mice with BLT images to determine the sagittal depth of the light source from the edge of the skull. As shown in Fig. 5, it was possible to determine the depth of cell implantation on day one with no statistically significant differences between CBR2opt/NH$_2$-NpLH2 (sagittal depth 3.0 mm ± 0.4 mm; $n = 3$) and CBR2/D-LH2 (sagittal depth 3.2 mm ± 0.5 mm; $n = 3$). Note data represent means ± standard deviation (S.D.). Videos are available as supporting material (Supplementary Movie 1 is CBR2opt with NH$_2$-NpLH2; Supplementary Movie 2 is CBR2opt with D-LH2). Moreover, we performed acquisition after 5 days to measure the possible migration of HEK-293 cells from their original location. In the case of CBR2opt/NH$_2$-NpLH2 we obtained highly resolved images of HEK-293 cells that had migrated to a different location from the 3D light reconstruction in the brain. As shown in Fig. 6a, reconstruction using CBR2/D-LH2 showed only a single larger spot, whereas CBR2opt/NH$_2$-NpLH2 reconstruction allowed us to locate two distinct signals in the brain (Fig. 6b). Histological analysis of coronal brain sections revealed the presence of copGFP positive cells located in both of the ventricular areas (calculated distance of ~ 1 mm), and confirmed the accuracy of reconstruction using CBR2opt/NH$_2$-NpLH2 (Fig. 6c).

## Discussion

In this study we report the design and characterization of two naphthyl-based luciferin analogs and the development of a mutant luciferase enzyme (CBR2) that can efficiently utilize these substrates to produce NIR bioluminescence. The amino compound (NH$_2$-NpLH2; peak emission at 730 nm) is of particular interest because of its demonstrated utility in mice. Prior to this report, the most red-shifted, in vivo-compatible bioluminescence system (without an energy transfer acceptor) was firefly luciferase combined with the substrate iLH2, which peaks at 706 nm[13].

Although longer wavelength signals are important for animal imaging, a luciferase/luciferin system must also produce ample photons for detection. In addition, substrates that are readily cell permeable are necessary for optimal biodistribution and sufficient production of luminescence[16, 30]. In consideration of these factors, we characterized cells expressing CBR2opt treated with D-LH2 and demonstrated that this combination produces higher photon flux compared to Luc2/D-LH2 and comparable flux to Luc2/Aka-HCl. The enhancement of CBR2opt over Luc2 (with D-LH2 as substrate) was likely a result of higher enzyme stability rather than enhanced catalytic efficiency. Moreover, the naphthyl-luciferin substrates retained their designed species specificity and were inefficient substrates for Luc2. Consequently, the naphthyl substrates (particularly NH$_2$-NpLH2) have the potential to selectively detect CBR/CBR2 in multicolor orthogonal BLI applications with different enzyme/substrate pairs (e.g., Luc2/caged D-LH2)[29, 31–35]. In addition, we showed that NH$_2$-NpLH2 has improved cell membrane permeability compared to D-LH2. Although encouraging, the cell-based results are not sufficient for the accurate prediction of in vivo behavior of a substrate, because parameters such as tissue biodistribution can significantly influence the sensitivity of bioluminescence systems.

For this reason we selected two challenging models to test our system: imaging transplanted cells in the backs of black (non-

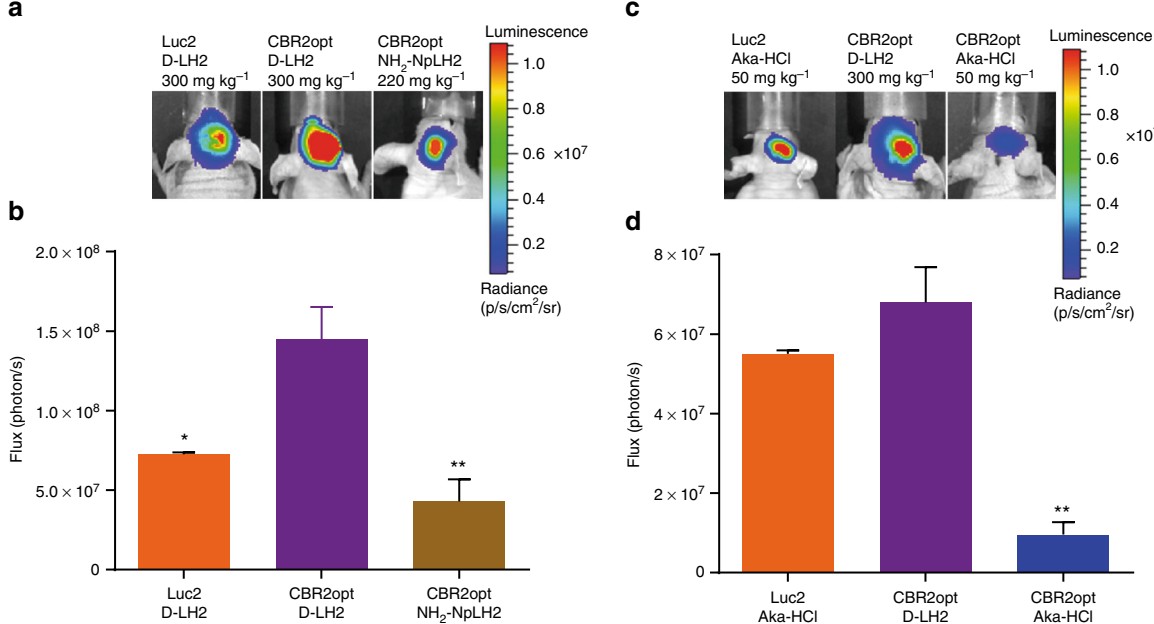

**Fig. 4** Comparison between CBR2opt and Luc2 utilizing either D-LH2 or Aka-HCl for imaging mouse brain. **a** Images of representative mice after intracranial implantation of $1 \times 10^5$ HEK-EF1-Luc2-T2A-copGFP or HEK-EF1-CBR2opt-T2A-copGFP cells and intravenous injection of D-LH2 (300 mg kg$^{-1}$) or NH$_2$-NpLH2 (220 mg kg$^{-1}$) ($n = 4$). **b** Quantification of photon flux (photon per s) with an exposure time of 30 s at the peak of bioluminescence emission (4–7 min after substrate injection) (*$p < 0.05$; **$p < 0.01$ compared to CBR2opt/D-LH2). **c** Images of representative mice after intracranial implantation of cells and intravenous injection of either D-LH2 (300 mg kg$^{-1}$) or Aka-HCl (50 mg kg$^{-1}$) ($n = 3$). **d** Quantification of flux (photon/s) with an exposure time of 30 s (**$p < 0.01$ compared to CBR2opt/D-LH2). Error bars represent S.D.

shaved) mice, and imaging deeper regions of mouse brain. When equivalent amounts of substrates were injected intraperitoneally in black furred mice, CBR2opt/D-LH2 produced more signal compared to both Luc2/D-LH2 and CBR2opt/NH$_2$-NpLH2. For comparisons of enzymes and substrates in the brain, we selected the optimal concentration of each substrate for relevant benchmarking. Luc2/Aka-HCl was included in the brain experiments because this pairing was recently reported to provide improved sensitivity over Luc2/D-LH2 in mouse lung when low, equimolar concentrations of substrate were used[16].

While comparing substrates under these conditions is scientifically valid, it is arguably more relevant to benchmark each system using optimal conditions. At peak signal intensity, CBR2opt/D-LH2 and Luc2/Aka-HCl produced comparable signals in deep brain, and both were superior to Luc2/D-LH2 and CBR2opt/NH$_2$-NpLH2. This demonstrated that the total light emission and the optimization of imaging conditions and parameters play an important role in the ultimate sensitivity for BLI in vivo. For specific applications in brain that do not require the high resolution provided by NIR-emitting substrates, alternative D-luciferin analogs that produce increased signal have recently been described[36].

The reason CBR2opt/NH$_2$-NpLH2 was not as sensitive as CBR2opt/D-LH2 or Luc2/Aka-HCl is likely that it did not produce as many photons > 620 nm. However, we achieved on average higher photon flux in vivo using CBR2opt/NH$_2$-NpLH2 compared to a recently described NIR bioluminescence system based on intramolecular BRET (RLuc8-iRFP720 fusion)[37]. Note that CBR2opt/NH$_2$-NpLH2 can potentially be combined for dual luciferase applications in vivo with a variety of BRET-based systems, e.g., iRFP720-Rluc8 and Antares (NLuc[38, 39]-CyOFP1 fusion)[40], as these utilize coelenterazine-based substrates.

Moreover, the NH$_2$-NpLH2 substrate did provide improved BLI spatial resolution. This was likely a result of reduced signal scattering associated with the longer wavelength photons. To

explore this possibility, we examined CBR2opt/NH$_2$-NpLH2 in the brains of mice using BLT and compared to CBR2opt/D-LH2. Reconstruction of single light sources was not significantly different using the two substrates. However, we were able, for the first time, to visualize at high resolution the migration of cells using CBR2opt/NH$_2$-NpLH2. This enzyme/substrate combination facilitated the use of multispectral acquisitions that are required to perform bioluminescence source reconstruction[41, 42]. Although an absolute quantification of resolution and accuracy of BLT was beyond the purpose of this study, we believe that the CBR2opt/NH$_2$-NpLH2 system can serve as a model for developing an improved algorithm for BLT[42, 43].

In summary, the engineered CBR2opt gene sequence in combination with D-LH2 or NH$_2$-NpLH2 enabled both sensitive and highly resolved imaging in vivo. CBR2opt/D-LH2 is a practical choice for general imaging applications, as it provides high sensitivity and utilizes a well understood and readily available substrate. For more specialized applications, e.g., tomography, where high resolution is important, the combination of CBR2opt with NH$_2$-NpLH2 is a valuable alternative. The versatility of this mutant enzyme paired with either D-LH2 or NH$_2$-NpLH2 is of particular interest and may eventually lead to the broad use of these systems for a variety of BLI applications in the future.

## Methods
Synthesis of NIR naphthyl-luciferin substrates (see Supplementary Methods)

**Computational molecular modeling**. Molecular modeling, analyses, and visualizations were performed using Discovery Studio (Dassault Systemes Biovia, formerly Accelrys Software).

The homology model of CBR was created with Modeler as implemented in Discovery Studio using default parameters[44]. The template structure was PDB code 2D1S[23] and the active site-bound analog, DLSA, was included in the model (DLSA = 5′-O-[N-(dehydroluciferyl)-sulfamoyl] adenosine)[45, 46]. In our CBR model, we replaced the dehydroluciferin moiety of DLSA with OH-NpLH2. To resolve bump contacts and relax the structure, we energy minimized the ligand plus CBR side

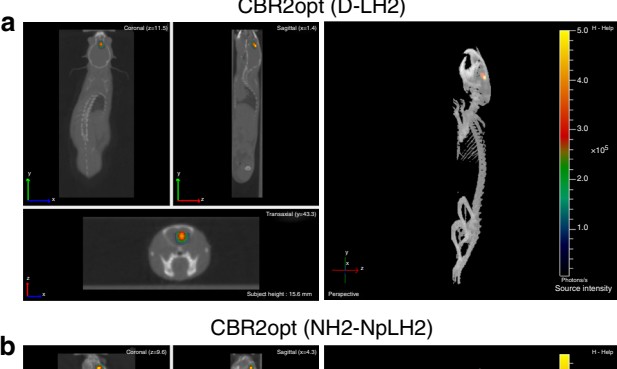

CBR2opt (D-LH2)

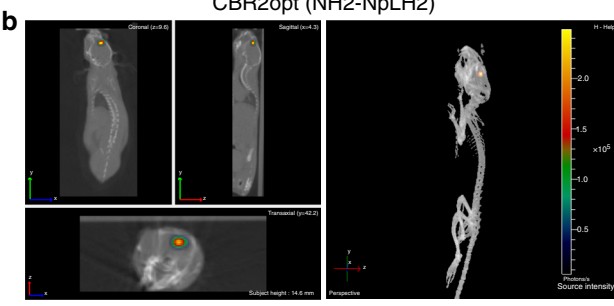

CBR2opt (NH2-NpLH2)

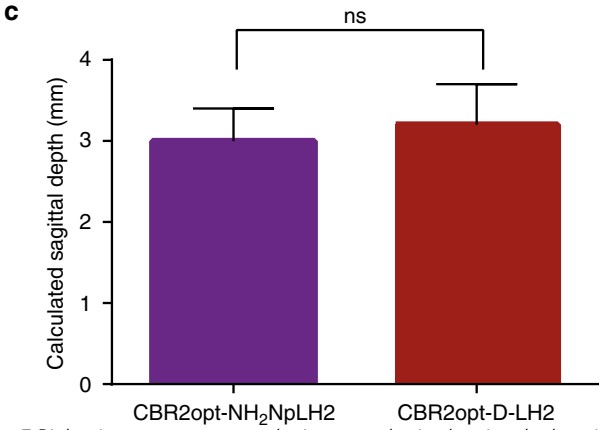

**Fig. 5** Bioluminescence tomography in mouse brain showing the location of the bioluminescence signal in mouse brain at day 1 after transplantation of cells. The images were acquired 10 min after intraperitoneal injection of NH$_2$-NpLH2 (220 mg kg$^{-1}$) or D-LH2 (300 mg kg$^{-1}$) using respectively band pass filters 700 nm, 720 nm, and 740 nm or 580 nm, 600 nm, and 620 nm for each mouse. Acquisition time was 30 s. After BLI mice kept in the dedicated coil were placed in the CT scanner. **a** Co-registered CT and BLT images using CBR2opt/D-LH2 pairing (sagittal section and reconstruction). **b** Co-registered CT and BLT images using CBR2opt/NH$_2$-NpLH2 pairing (sagittal section and reconstruction). **c** Quantification of depth from skull surface indicated no significant difference (ns) between substrates when using the single point light source ($n = 3$). Error bars represent S.D.

chains within 5 Å, with dihedral restraints on the ligand and a harmonic restraint on the peptide backbone. To better assess hydrogen-bonding patterns we then ran a molecular dynamics simulation using the Standard Dynamics Cascade protocol and the same restraints as above.

**Plasmid constructions.** CBR mutants/mutant libraries were constructed using QuikChange II Site-Directed Mutagenesis Kit (Agilent Technologies 200523) according to the manufacturer protocol. Oligonucleotides were from IDT. Mutagenesis reactions were used to transform E. coli KRX (Promega). Individual colonies were picked for plasmid preparation and DNA sequence verification. All plasmids for bacterial expression and transient mammalian cell expression were in a pF4Ag backbone (T7 and CMV promoters; Promega). For purification from bacterial overexpression, sequences were sub-cloned to pF6HisNK (Promega). CBR2opt was assembled synthetically as a mammalian codon optimized version of CBR2 expressing the identical enzyme sequence as CBR2 (Gene Dynamics). For

stable cell line generation, Luc2 and CBR2opt were sub-cloned from their pF4Ag backbones into lentiviral vector pCDH-EF1-MCS-T2A-copGFP (System Biosciences).

**Screening in bacterial lysates.** Individual colonies were added to 96-well plates containing LB media plus antibiotic. Plates were grown overnight at 37 °C. Induction media was inoculated with the overnight culture (1:20) and grown overnight at 25 °C in autoinduciton media (LB media plus antibiotic with 0.2% rhamnose and glucose) to induce protein expression. Cells were lysed using PLB (Promega) and then assayed with NH$_2$-NpLH2 or OH-NpLH2 in assay buffer (150 mM HEPES (pH 7.5), 1 mM CDTA, 16 mM MgSO$_4$, and 1% NP-9) containing 1 mM ATP. Luminescence was measured using an ImageQuant LAS400 CCD imager (GE Healthcare).

Note monopotassium salt forms of each substrate were used for all characterizations.

**Protein purification.** The CBR/CBR2 enzymes were prepared by overexpression in KRX E. coli and then isolating to > 90% purity using MagneHis™ Protein Purification System (Promega) following the manufacturer protocol. The source of Luc2 luciferase was QuantiLum Recombinant Luciferase (Promega). The source of UltraGlo luciferase was Promega.

**$K_m$ and RLU$_{max}$ determination for D-Luciferin.** 10 mM ATP was added to assay buffer (150 mM HEPES (pH 7.5), 1 mM CDTA, 16 mM MgSO$_4$, and 1% NP-9) and this solution was used as a diluent to create two separate 3-fold serial dilution series of D-LH2 starting at 0.3 mM to 0.000066 mM and 1 mM to 0.00022 mM. Each dilution series was prepared in triplicate and Luc2, CBR, and CBR2 were diluted to 0.2 μg mL$^{-1}$ in TBS. In quadruplicate, 50 μL of each dilution series starting with 0.3 mM was combined with 50 μL of the Luc2 dilution, and 50 μL of the dilution series starting at 1 mM was combined with 50 μL of the CBR and CBR2 dilutions. Each plate of samples was incubated for 5 min at room temperature and then read on a GloMax®-Multi+ luminometer (Promega). $K_m$ and RLU$_{max}$ were calculated using GraphPad Prism Michaelis–Menten regression. Average and standard deviation for $K_m$ and RLU$_{max}$ were calculated from each triplicate dilution series. All luminescence values were converted to RLU μM$^{-1}$ of enzyme.

**$K_m$ and RLU$_{max}$ determination for NH$_2$-NpLH2.** 10 mM ATP was added to assay buffer (150 mM HEPES (pH 7.5), 1 mM CDTA, 16 mM MgSO$_4$, and 1% NP-9) and this solution was used as a diluent to create triplicate 3-fold dilution series starting at 20 μM NH$_2$-NpLH2. (20 μM to 200 μM) Luc2 was diluted to 200 μg mL$^{-1}$ in TBS and CBR and CBR2 were diluted to 20 μg mL$^{-1}$ TBS. In quadruplicate 50 μL of each substrate dilution series was combined with 50 μL enzyme dilution. Plates were incubated for 5 min and then read on a GloMax®-Multi+ luminometer (Promega). $K_m$ and RLU$_{max}$ were calculated using GraphPad Prism Michaelis-Menten regression. Average and standard deviation for $K_m$ and RLU$_{max}$ were calculated from each triplicate dilution series. All luminescence values were converted to RLU μM$^{-1}$ of enzyme. It was not possible to calculate accurate kinetic parameters for Luc2 because there was insufficient signal over background at low substrate concentrations.

**$K_m$ and RLU$_{max}$ determination for ATP.** Both 1 mM and 0.3 mM solutions of D-LH2 were prepared in assay buffer (150 mM HEPES (pH 7.5), 1 mM CDTA, 16 mM MgSO$_4$, and 1% NP-9) 6 mM ATP was added to an aliquot of each of the D-LH2 solutions. The remaining D-LH2 solutions were used as a diluent to prepare in triplicate 2-fold serial dilutions of ATP (6 mM to 0.047 mM). 50 μL of each ATP dilution series was then added to 50 μL of each diluted enzyme solution. Plates were incubated for 5 min and then read on a GloMax®-Multi+ luminometer (Promega). $K_m$ and RLU$_{max}$ were calculated using GraphPad Prism Michaelis-Menten regression. Average and standard deviation for $K_m$ and RLU$_{max}$ were calculated for each triplicate dilution series. All luminescence values were converted to RLU μM$^{-1}$ of enzyme.

**Thermal stability.** HEK 293 cells (ATCC® CRL-1573™) identified by STR analysis were cultured in DMEM, 10% FCS and additional penicillin/streptomycin antibiotics. Cells were negative for mycoplasma contamination. HEK 293 cells expressing Luc2, CBR, and CBR2 were lysed with 1X PLB. Lysates were transferred into a 96 well PCR tray and then incubated in a Veritas thermal cycler (ABI) at 37 °C. At various time points, lysate was removed from heat treatment and stored at 4 °C. After all of the samples had been transferred, samples were equilibrated to room temperature and then assayed in triplicate with assay buffer (150 mM HEPES (pH 7.5), 1 mM CDTA, 16 mM MgSO$_4$, and 1% NP-9) supplemented with 1 mM D-LH2 and 1 mM ATP. Half-life values were calculated using Graphpad Prism one phase decay regression (plateau set to zero).

**Spectral properties.** Purified preparations of Luc2, CBR, and CBR2 were diluted to 50 μg mL$^{-1}$ in TBS pH 7.5 + 0.01% BSA and then 10-fold serial dilutions were prepared in TBS. Substrates were diluted into assay buffer (150 mM HEPES (pH 7.5), 1 mM CDTA, 16 mM MgSO$_4$, and 1% NP-9) (0.5 mM D-LH2, 0.02 mM NH$_2$-

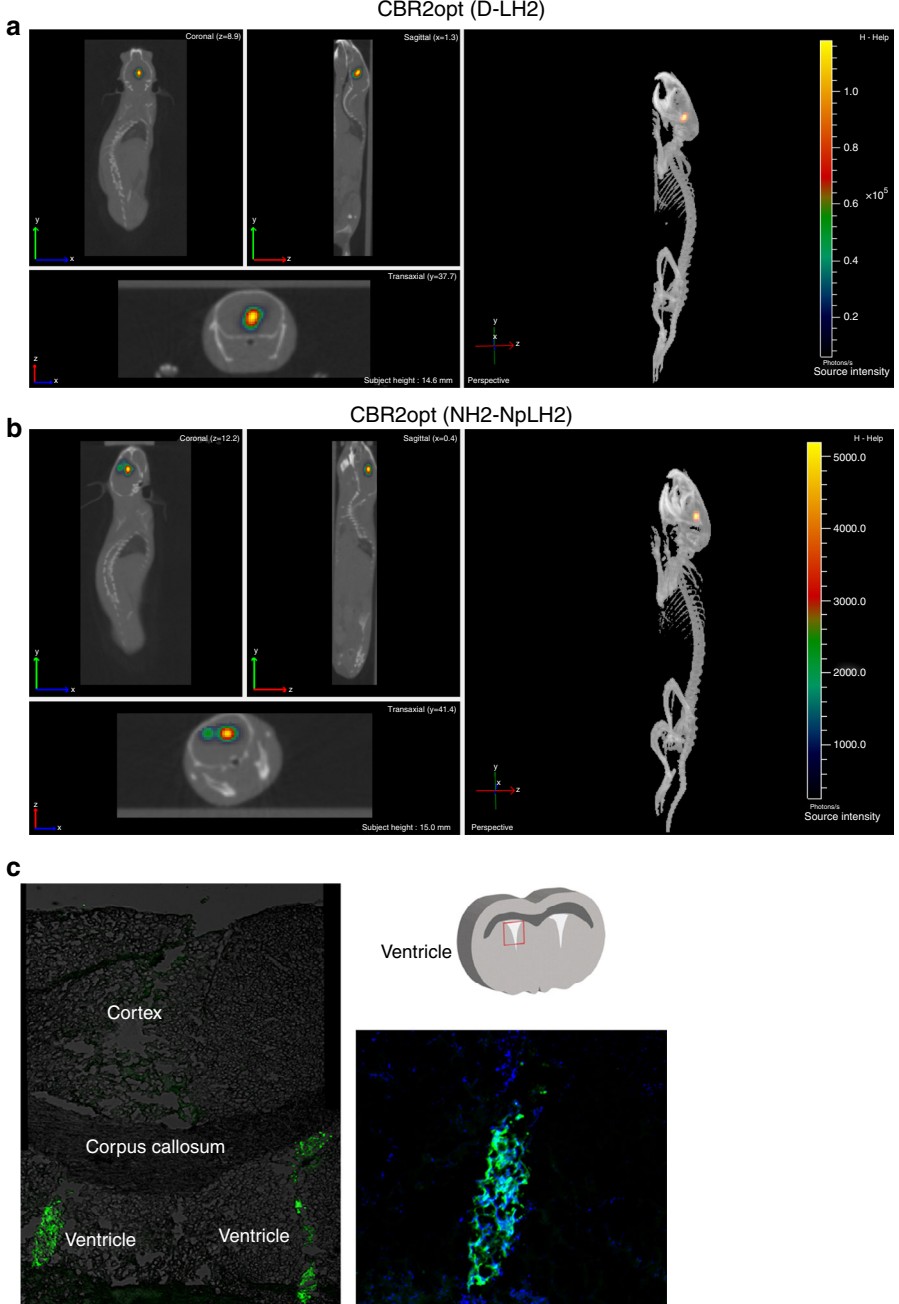

**Fig. 6** Bioluminescence tomography in mouse brain showing the location of the bioluminescence signal at day 5 after transplantation of cells. The images were acquired after intraperitoneal injection of NH2-NpLH2 (220 mg kg$^{-1}$) or D-LH2 (300 mg kg$^{-1}$) using respectively band pass filters 700 nm, 720 nm, and 740 nm or 580 nm, 600 nm, and 620 nm for each mouse. Acquisition time was 30 s. After BLI mice kept in the dedicated coil were placed in the CT scanner. **a** Coronal, transaxial and sagittal view of co-registered CT and BLT images using CBR2opt/D-LH2 pairing. **b** Coronal, transaxial and sagittal view of co-registered CT and BLT images using the CBR2opt/NH2-NpLH2 pair. Migration of cells can be clearly seen at day 5 (two adjacent light sources are represented). **c** Histological analysis showing the presence of two groups of cells (copGFP) in both ventricular areas of brain (green signal = copGFP; blue = nuclei). Scale bars = 500 μm (left panel) and 10 μm (right panel)

NpLH2, or 0.04 mM OH-NpLH2) containing 10 mM ATP. In triplicate, 50 μL of substrate dilution and 50 μL of enzyme dilution were combined and immediately measured on a Tecan Infinite® M1000 plate reader set to spectral scanning mode with 2 nm intervals (500–850 nm). The following enzyme dilutions were used for D-LH2: 0.05 μg mL$^{-1}$ Luc2 or 0.5 μg mL$^{-1}$ CBR/CBR2. The following dilutions were used for NH2-NpLH2 and OH-NpLH2: 50 μg mL$^{-1}$ for Luc2 or 5 μg mL$^{-1}$ CBR/CBR2.

Spectral measurements on live cells were carried out as follows: cells were transfected with Luc2 and CBR2opt as described in the "Transfections" section below. Substrates were pre-heated to 37 °C and then 30 μL of 100 mM D-LH2 and 30 μL of 4.5 mM NH2-NpLH2 were each added to three wells of Luc2 and CBR2opt-transfected cells. The plate was manually shaken and immediately

transferred to a Tecan Infinite® M1000 plate reader heated to 37 °C. The plate was incubated for 5 min and then measured in spectral scanning mode with 3 nm intervals (500–850 nm).

**Cell preparation for live cell assays**. Growth media (DMEM, Life Technologies) supplemented with 10% FBS (Seradigm) was aspirated from a confluent flask of HEK-293 cells and then adherent cells were washed with DPBS (Life Technologies 14190). DPBS was aspirated and the cells were released from the flask by the addition of 3 mL of TrypLE™ Express Trypsin (Life Technologies) and incubation at 37 °C. Cells were centrifuged at 300 x g washed, re-suspended in 10 mL of fresh growth media, and then counted using a BioRad TC20 cell counter. Cells were

diluted to a concentration of 200,000 cells per mL and then 3 mL of cells were dispensed into 6-well tissue culture plates (Corning 3506) at 600,000 cells/well. Plates were grown overnight at 37 °C with CO2.

**Transfections**. Plasmid DNA (Plasmid.com™) from each luciferase clone (Luc2, CBR, CBR2, CBR2opt) was diluted to a concentration of 0.02 μg μL$^{-1}$ in a volume of 465 μL of Opti-MEM (Life Technologies) and then 30 μL of FuGENE® HD Transfection Reagent (Promega) was added to each diluted DNA. Transfection complexes were incubated for 10 min at ambient temperature and then 150 μL of each sample was added to three wells of the previously plated HEK 293 cells. Plates were manually mixed after complex addition and then incubated at 37 °C with CO2 for 20 h. Growth media was aspirated from each well of the 6-well plates and washed with DPBS. After DPBS aspiration, cells were released by the addition of 1 mL of Triple™ Express Trypsin and incubated at 37 °C. Fresh growth media was added to each well and triplicate transfection reactions for each sample were pooled, washed, counted, and then diluted to a concentration of 200,000 cells per mL. Three white plates (Costar 3917) and three black plates (Costar 3916) were filled with 100 μL of cells for each transfection reaction (24 wells per plate for each sample.) Plates were incubated at 37 °C with CO2 for 20 h.

**$K_m$ and RLU$_{max}$ determination in live cells**. The following substrate solutions were prepared in DPBP: 100 mM D-LH2 (Promega), 8 mM of NH2-NpLH2, and 8 mM of OH-NpLH2. Two-fold serial dilutions were prepared for each substrate. (100 mM to 0.78 mM or 8 mM to 0.0625 mM). 30 μL of the D-LH2 titration was added to cells transfected with either Luc2, CBR, CBR2, or CBR2opt in a white plate. The plate was manually shaken, and immediately placed in a GloMax®-Multi + luminometer (Promega) set to 37 °C. The plate was then read at 3 and 10 min after substrate addition. This procedure was repeated for NH2-NpLH2 and OH-NpLH2. The 3 min time point was used for $K_m$ and RLU$_{max}$ calculations. From the same set of dilution series, 30 μL of each substrate dilution was added to the transfected cells in the black plates. Black plates were manually shaken and luminescence was read on the ImageQuant LAS4000 CCD imager (GE) at ambient temperature. The D-LH2 plate was read using a 30 s exposure, the NH2-NpLH2 plate was read using a 600 s exposure, and the OH-NpLH2 plate was read using a 2000 s exposure. $K_m$ and RLU$_{max}$ values were calculated using GraphPad Prism (Michaelis–Menten regression).

**Cell-membrane permeability assays**. HEK-293 cells plated to a density of 600,000 cells in a 6-well plate were grown overnight. The following day cells were transfected with 10 μg of either Luc2 or CBR2opt in the presence of 60 μL of Fugene (Promega). Cells were grown overnight and then 100 μL of each transfected cells were re-plated into 96-well white assay plates at a concentration of 20,000 cells per well and then grown overnight. As a means to determine permeability, 20 μL of TBS (no lysis) or 20 μL of lysis reagent (either 5X PLB (Promega), or 500 μg mL$^{-1}$ digitonin) was added to cells and they were allowed to shake for 10 min at 600 rpm. Assay reagents containing 250 μM of either D-LH2 or NH2-NpLH2 and 50 μM ATP were added (12 μL) to the TBS treated cells and the PLB or digitonin treated cells (25 μM final substrate, 5 μM final ATP). Plates were shaken for 1 min and then luminescence was measured on a GloMax®-Multi+ luminometer (Promega). The ratio of luminescence from lysed cells to un-lysed cells was calculated.

**Preparation of lentivirus**. Lentivirus was produced, as previously described[47]. Briefly, lentiviral particles were produced by transfection of HEK-293 T packaging cells with three packaging plasmids (pCMV-VSVG, pMDLg-RRE (gag-pol), pRSV-REV; Addgene) and the lentiviral vector plasmid using PEI transfection reagent (1 mg mL$^{-1}$) per μg DNA). Supernatants containing lentiviral particles were collected after 48 and 72 h. Subsequent quantification of virus was performed using a standard antigen-capture HIV p24 ELISA (ZeptoMetrix).

**Transduction of cells and selection for equimolar expression**. HEK-293 cells were cultured in DMEM with addition of 10% FBS, penicillin and streptomycin. Cells were seeded in a 24 well plate at a density of 75,000 cells/well and transduced with MOI 1 of either EF1-Luc2-T2A-copGFP or EF1-CBR2opt-T2A-copGFP lentivirus plus polybrene (hexametridine bromide, Sigma) at a final concentration of 8 μg mL$^{-1}$. Cells were subsequently passaged and sorted two times for GFP expression using FACS (BD-FACS AriaIII, BD Biosciences).

**Fluorescence imaging**. Transduced HEK-EF1-Luc2-T2A-copGFP or HEK-EF1-CBR2opt-T2A-copGFP were seeded in a 96-well black plate at a density of 50,000 cells per well and left to adhere. Subsequently, cells were imaged using a fluorescence microscope (Leica Microsystems) for expression of GFP.

**In cell western analysis**. Transduced HEK-EF1-Luc2-T2A-copGFP or HEK-EF1-CBR2opt-T2A-copGFP were seeded in a 96-well black plate at a density of 50,000 cells/well, left to adhere, washed and fixed using 3.7% formaldehyde in PBS for 15 min, and then treated with 0.1% saponinin PBS for 10 min at ambient temperature. Wells were rinsed three times for 5 min each with ambient temperature PBS (100 μL per well). Cells were then blocked in Blocking Buffer (LI-COR Biosciences) for

1 h at ambient temperature (50 μL per well). Blocking Buffer was removed and then buffer (negative control) or 1:1,000 rabbit polyclonal anti-TurboGFP antibody (Evrogen AB513) was added to the wells (total volume 50 μL per well). The plate was covered and incubated overnight at 4 °C. The next day, cells were washed 3× in PBS and then anti-Rabbit IgG (H+L) (DyLight® 800 Conjugate) secondary antibody (diluted in Antibody Dilution Buffer (LI-COR Biosciences), (total volume 50 μL per well)) was added. After a 1 h incubation at ambient temperature in the dark, cells were washed 3× with PBS and scanned using an Odyssey scanner (LI-COR Biosciences) using the following settings: filter 800, intensity 5, and resolution 42 μm.

**Live cell imaging stable cell lines**. HEK-EF1-Luc2-T2A-copGFP or HEK-EF1-CBR2opt-T2A-copGFP were seeded in a 96-well black plate at a density of 10,000 cells per well. After 24 h cells were washed in PBS and imaged after addition of substrate (D-LH2, NH2-NpLH2, OH-NpLH2 (potassium salts) and Aka-HCl (Toke-Oni, Sigma) diluted in medium at a final concentration of 1 mM (100 μL per well). For the determination of flux (photon/s/cell) a range of $5 \times 10^4$, $2.5 \times 10^4$, $1.25 \times 10^4$, $6.12 \times 10^3$, $3.125 \times 10^3$, and $1.562 \times 10^3$. HEK-EF1-Luc2-T2A-copGFP or HEK-EF1-CBR2opt-T2A-copGFP cells were seeded in a black 96-well plate with a clear bottom and imaged after addition of 1 mM D-LH2. Cells were imaged using an IVIS Spectrum (Perkin Elmer) 5 min after substrates addition using the following setting: FOV C, medium binning and 30 s acquisition with open filter. Experiments were performed in sextuplicate and repeated twice. Data were analyzed using Living Image 4.3 software (Perkin Elmer) by drawing the appropriate region of interest (ROI) and then plotted using Graphpad Prism.

MCF7 cells (ATCC HTB-22) were thawed and cultured in DMEM plus 10% of FBS and additional penicillin and streptomycin. Cells were seeded in a black 96-well plate with clear bottom at a concentration of 20,000 cells/well and transfected the subsequent day using 0.11 μg of plasmid DNA per well for the expression of Luc2 or CBR2opt and 0.3 μL of FuGENE® HD reagent/well. After 24 h cells in the plate were washed in PBS and imaged after addition of luciferase substrates (D-LH2 potassium salt, NH2-NpLH2, OH-NpLH2 and Aka-HCl) diluted in medium at a final concentration of 1 mM. Cells were imaged using IVIS Spectrum 5 min after substrates addition using the following setting: FOVC, medium binning and 30 s acquisition with open filter. Experiments were performed in sextuplicate and repeated twice. Data were analyzed using Living Image 4.3 software (Perkin Elmer) by drawing the appropriate ROI and then plotted using Graphpad Prism.

**In vivo bioluminescence imaging**. Animal experiments were reviewed and approved by the Bioethics Committee of Leiden University, The Netherlands. Animal care and handling was in accordance with the guidelines and regulations as stipulated by the Dutch Experiments on Animals Act (WoD) and the European Directive on the Protection of Animals Used for Scientific Purposes (2010/63/EU).

For the subcutaneous skin model experiments, 8–10 week old C57BL/6 black fur mice ($n = 3$ per group) received a subcutaneous injection of $1 \times 10^6$ HEK-EF1-Luc2-T2A-copGFP and HEK-EF1-CBR2opt-T2A-copGFP cells (resuspended in 30 μL) and a subsequent injection of a standard dose (150 mg kg$^{-1}$) of D-LH2 or NH2-NpLH2.

For experiments involving deep mouse brain, on day 1, 6–8 week old CD-1 nude mice were anesthetized using isofluorane and placed in a robot stereotactic device (Neurostar). $1 \times 10^5$ HEK-EF1-Luc2-T2A-copGFP or HEK-EF1-CBR2opt-T2A-copGFP cells were counted using an automated cell counter, prepared in PBS solution at a concentration of $5 \times 10^7$ cells per mL and analyzed for GFP expression. Skulls were drilled and cells were subsequently injected in a volume of 2 μL into the striatum (coordinates relative to bregma: AP + 0.5; L + 2.0; DV −3.0). We expected a difference in photon flux mean values of >50%. Therefore, groups of 3 or 4 mice were used for every condition. On day 2 mice received an intravenous injection (200 μL) of D-LH2 (300 mg kg$^{-1}$), NH2-NpLH2 (220 mg kg$^{-1}$) or Aka-HCl (50 mg kg$^{-1}$). Cell preparation was carried out by one scientist, and transplantations and substrate administration were performed blindly by a second scientist. Acquisition of images was performed by a third scientist.

Dosing of substrates was based on maximum solubility, maximum attainable signal and tolerability in mice based on previous findings. Mice were kept under isofluorane anesthesia (1.5%) and a series of images were taken using an IVIS Spectrum using open filter binning=medium, field of view=12.9 × 12.9 cm, $f$/stop = 1 and either 30 or 60 s exposure time for cells transplanted in the brain or subcutaneously, respectively. Data analysis was performed by drawing ROIs in the images taken at the peak of bioluminescence emission.

**Bioluminescence tomography in brain**. For BLI tomography of cells transplanted in mice brain, CBR2 expressing cells were transplanted as described above at an approximate depth of 3 mm from the dura in 8 week-old BALB/c mice ($n = 6$). D-LH2 (300 mg kg$^{-1}$) or NH2-NpLH2 (220 mg kg$^{-1}$) was injected intraperitoneally 1 and 5 days later. For comparing bioluminescence tomography between different substrates, NH2-NpLH2 was administered 3 h after D-LH2 (i.e., when the signal from D-LH2 had dropped to background). Signals were collected using different bioluminescence filters on an IVIS Spectrum system 15 min after injection. The following conditions were used for imaging acquisition: exposure time 30 s, binning ¼medium: 8, field of view 12.9 cm and f/stop 1. Band pass filters 580, 600, and 620

nm were used for mice receiving D-LH2 and band pass filters 700, 720, and 740 nm for mice receiving NH₂-NpLH2. Mice were kept inside of an appropriate coil for subsequent CT scanning using a QuantumFX scanner (Perkin Elmer). Reconstruction of 3D bioluminescence and co-registration with CT images were performed using Living Image 4.5 software (PerkinElmer).

**Histological analysis**. At day 5 post cell injection mice were killed and perfused using 4% paraformaldehyde (PFA). Brains were dissected and cryopreserved for subsequent cutting using a cryostat. Coronal sections of 20 μm were placed in microscope glass slides, fixed with 1% PFA, washed and stained with DAPI for nuclear staining. After washing with PBS and addition of mounting media, cover glasses were section screened using a fluorescent microscope (Leica Microsystems). When cells were localized multiple brightfield images and copGFP fluorescence images were stitched together to reconstruct a partial brain structure.

**Statistical analysis**. Data are presented as means ± standard deviation (S.D.) and where appropriate analyzed using ONE-way ANOVA followed by a Tukey's *t* test for multiple column comparison. *P* values < 0.05 were considered to be statistically significant.

**Data availability**. Authors confirm that all relevant data are included in the paper and/or its supplementary information files. Other data that support the findings of this study are available from the corresponding author on request.

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

## Acknowledgements

We thank Kris Zimmerman and Gregg Colwell (Gene Dynamics) for help with vector constructions, Paul Otto for help with enzyme purifications, Laurent Bernad for help with analytical chemistry, Andrew Niles for help with substrate toxicity testing, and Dieter Klaubert for early discussions about the design of naphthyl-luciferins. We also acknowledge Dr. Luis Cruz for help with lab equipment, and thank Lancy Iljas for technical help with cell preparation for experiments and Timo Schomann for technical help with the microscope. This work was supported by the FP7 European Union Marie Curie IAPP Program, BRAINPATH, under grant number 612360, European H2020 MSCA award under proposal number 675743 (project acronym: ISPIC) and the Applied Molecular Imaging Erasmus MC (AMIE) facility.

## Author contributions

M.P.H., C.C.W., L.M., C.S., and L.P.E. designed and performed experiments. I.Q., M.R., and Y.R. provided technical assistance for in vivo imaging experiments. C.C.W., C.S., M. W., and M.P.H. designed, developed and characterized the substrates. M.P.H., L.P.E., and L.M. wrote the manuscript. M.P.H., C.C.W., M.G.W., C.S., T.A.K., L.P.E., K.V.W., C.L., and L.M. critically reviewed/revised the manuscript.

## Additional information

**Competing interests:** The authors declare no competing financial interests.

