## [Peer Review File · Nature Communications]

Reviewers' comments:

Reviewer #1 (Remarks to the Author):

I am providing a review of the manuscript “Novel click beetle luciferase mutant and near infrared naphthyl-luciferins for improved bioluminescence imaging”. This manuscript contains some very interesting and potentially highly significant results that will appeal to a broad readership because of the importance of bioluminescence-based imaging of live animals. The authors address problems associated with deep tissue imaging that include the need for bioluminescence sources that produce sufficient photon flux at wavelengths greater than ~620 nm. A major contribution is transforming a commercially successful luciferase (CBR) into a variant (CBR2opt) that is mammalian codon optimized and contains 2 point mutations discovered by rationally design using molecular graphics. Additionally, 2 new luciferin analogues are reported that contain an additional benzene ring inserted between the 2 rings of the benzothiazole moiety of the natural substrate beetle luciferin. The extended conjugation of the amino and hydroxyl-substituted substrates results in impressive long wavelength maximal emission that is mainly in the near-infrared. In vitro testing results are presented along with figures that illustrate very nice imaging results in mice. An important result that is clearly documented in Figure 3 is the greater luminescence (the authors should state how much greater) intensity produced in black fur mice by the new luciferase CBR2opt with beetle luciferin compared to the more commonly used Luc2 protein. The brain imaging data likewise show the CBR2opt enzyme to be superior. It seems from Figure 4 that the new system is modestly better than Luc2 in brain imaging than with another commercial substrate called Aka-HCl. While substantial advantages of the 2 new substrates were not really obtained, it is very promising that the greater nearIR emission of one of the compounds will be advantageous for brain imaging by increasing resolution with acceptable sensitivity. Overall, the findings presented in the manuscript are very well documented, interesting, and important; they definitely warrant publication.

The manuscript, however, requires revision and tighter focus on the major findings. In general, while it is interesting to compare the in vitro and imaging results, it is somewhat confusing, in part, because of the differences in sensitivities of the two types of detection devices that were used. While the authors have presented some very well done comparative studies, one wonders why the brain imaging experiment illustrated in Figure 5 was done only with CBR2opt and one new substrate analogue. If the point was to demonstrate the superiority of this pair of reagents, it would be clearer if there were comparative data (for example with Aka-HCl/Luc2).

Some additional suggestions for revisions include the following.

1. State the DNA sequence of CBR2opt and compare it to CBR. What is different about the codon optimization?
2. Throughout the text vague comparative terms are used and should be substituted with more quantitative expressions. Examples are: substantially more, significantly more, signals were weak, extended period of time, lower than, higher affinity, not as significant, etc.
3. Though no standard abbreviations exist for luciferin analogues, NpLH2 has already been used to describe an analogue in which the naphthyl ring replaces the benzothiazole ring of luciferin. The authors should consider using an abbreviation that indicates their substrates contain a naphtho[2,1]thiazole ring system.
4. The authors state that the luciferins were made with high enantiopurity, but there is no experimental evidence presented to back up this claim. They should state the evidence.
5. On page 5, the word “theory” should be replaced with “hypothesis” or an equivalent.
6. On page 11, DLSA is described as a “high-energy intermediate”. It is not clear what is meant. DLSA is a potent inhibitor of luciferase.
7. Most journals require depositing the NMR spectra of new compounds.
8. In the Figure legends, “N=3” is used. Is this a standard abbreviation?

Reviewer #2 (Remarks to the Author):

I believe the researchers were trying to engineer a luciferase mutant to enhance the bioluminescence of two new naphthyl derived luciferins that showed nrIR bioluminescence with the CBR luciferase, but were, as usual, less bright than the luciferin/firefly luciferase system. The mutant luciferase was successfully developed by point mutations derived from analysis of modelling one of their naphthyl luciferin analogues in the active site of a firefly luciferase. The new mutant luciferase gave more light output with D-luciferin than it did with the amino-naphthalene luciferin analogue or the D-luciferin/firefly luciferase system in transplanted cells in the backs of black mice. The D-luciferin/mutant luciferase system also gave more light in their brain imaging model, but similar to the light emitted with Aka-HCL/firefly luciferase. The authors show that the unique combination of amino-naphthalene luciferin analogue/mutant

luciferase gives high enough intensity of nrIR light to enable bioluminescence tomography of infected cells in a mice brain model. I think the data fully supports the claims made in the manuscript.

The authors claim that their two new analogues are the most red shifted to date. Anderson has synthesized an extended version of their original infraluciferin which gives a weak, but real signal ~800 nm (RSC Advances 2017, 7, 3975). The authors state in their opening paragraph the need for sustained imaging in challenging tissues such as brain. I do not know why the authors have not commented on the recent work of Wu et al (Anal. Chem. 2017, 89, 4808) who have found that their N-cyclobutylaminoluciferins gives longer bioluminescent signals and 20-fold more light than D-luciferin at low concentrations in a mouse brain model. How do these results compare to the present study and surely they detract from the novelty of the claim? The fact that the new mutant luciferase is more stable than firefly luciferase and hence gives more light output is interesting, but I think an incremental advance in the mutation of luciferases, rather than novel. The bioluminescence tomography results are compelling, but to a non expert how do they compare to other non invasive methods of viewing similar brain cancers? I am not an expert in this area, but it would be nice if the results were set in context, especially for the broad readership of Nature.

Overall this is a large body of work that has been carefully done. I do not think it turned out as the authors had intended. Ideally the new luciferase mutant should have given superior results with the analogues than with D-luciferin. However, I do not think brightness should be the sole indicator to the value of nrIR luciferins. Because of their long wavelength emission they can give superior data in tissue, as demonstrated in this work. Overall though, I am more of the opinion that this work represents some very good incremental advances, as opposed to the successful demonstration of some genuinely novel bioluminescent systems. I therefore do not support the publication of this manuscript in Nature Communications.

Other comments.

Line 76. The weak signal strength of nrIR luciferin analogues is continually stated as a reason that limits their utility. I do not agree and think the authors should quantify and qualify this statement with examples and references.

Line 80. The authors should state 'however' that the relative enhancement of light from Aka-HCl was only evident when low concentrations of substrate were used. Is this necessarily deleterious as the authors are suggesting? Please elaborate or reword.

Line 107-110 is a very interesting point pertaining to the design of their analogues. Could the authors provide some references.

Line 112-113, a few sentences here to describe the chemical synthesis. There does not seem to be any evidence of enantiopurity (see later).

Line 113-114 what salt forms were made and how?

Line 125 55 nm not 65 nm longer, but comment not valid in the light of RSC Advances 2017, 7, 3975.

Line 128-129. This phrase about PMT bias crops up all over the manuscript. It needs to be

qualified and quantified if possible. It may be a limitation of the technique or machine, but it will be common to all.

Line 144-145. The OH analogue was modelled, so why are the authors commenting about the importance of H-bonds to amino luciferins. Surely it would have been better to model the NH₂ analogue. Unless of course this is how the research was conducted pedagogically, in which case explain.

Line 161-162. Again quantify PMT bias.

Pg 291. I do not understand this statement and it should be made more clear.

I am not qualified to judge the computational or biological techniques. Hopefully one of the other referees will be. I think there are some deficiencies in the chemical synthesis methods which would make it very difficult to be repeated by a practicing organic chemist.

Line 570 I think S2 and S3 should be presented in a different orientation to allow an easier of understanding of which functional groups have been transformed etc.

Line 571 and 574. Conventionally in this field the carboxyl stereocentre is drawn with stereochemistry, not part of the ring.

Yields should be included in the schemes and the optical purity.

Volumes of solvents need to be given throughout.

The new structures need to be fully characterized with ¹³C NMR and Mass Spec (and submitted as supplementary information), otherwise I cannot fully judge the compounds integrity or purity.

All NMR's need J values reporting for multiplicities.

The mass of S2 obtained, its percentage yield and melting point need to be stated.

S3 the isolated form needs to be stated and if a solid the melting point also.

S4 percentage yield and melting point should be stated.

NH₂-NpLH₂

Line 621 isolated form needs to be stated, possible melting point and full characterisation data.

The optical rotation needs to be stated. As does the optical purity, how it was determined and that evidence submitted as supplementary material.

S6 percentage yield and melting point needs to be stated.

S7 form of final product, possible melting point and percentage yield missing.

S8 what form was isolated, possible melting point, mass isolated and percentage yield all missing

OH-NpLH₂

Line 653 7-methoxy (?)

Line 655 How was neutralization measured.

What form was isolated, possible melting point, mass isolated and percentage yield all missing.

The optical rotation needs to be stated. As does the optical purity, how it was determined and that evidence submitted as supplementary material.

Prof JC Anderson 230617

END

Reviewer #3 (Remarks to the Author):

The present new analogs of D-luciferin and a mutant derivative of click beetle red luciferase to produce red-shifted light emission for bioluminescence imaging. Bioluminescence imaging remains the most commonly used small animal imaging modality for pre-clinical studies. Developing enzyme-substrate pairs with improved NIR emission characteristics represents an important advance in this area. The manuscript is written clearly with methods that others in the field could reproduce. Specific comments are listed below.

- 1) Several of the graphs lack statistical comparisons to match differences noted in the text. The authors should provide appropriate statistics and note significant and non-significant differences consistently.
- 2) The authors note no difference in detection of brain lesions following i.p. injection of D-LH2 or NH2-NP-LH2, yet they used i.p. administration for tomography in Figure 5. The authors should clarify decision-making for selecting a route of administration.
- 3) Data comparing different substrates in vivo are presented as photon flux values without any normalization to numbers of cells or tumor size. Since differences in Figure 4 are less than 2-fold, some type of normalization is needed to account for potential differences due to numbers of cells present in each animal.
- 4) For figure 5, the authors only show data for CBR2opt with NH2-NP-LH2. Two comments regarding this figure: 1) Comparison imaging with CBR2opt and D-LH2 is needed to justify use of the modified substrate to achieve better resolution for tomographic imaging; and 2) the authors should provide supporting evidence (such as histology or MRI) for the number of brain lesions present in these mice to determine sensitivity/specificity for detecting discrete tumor foci with bioluminescence.

Reviewer #1 response:

Reviewer #1 clearly understands the significance of bioluminescence for small animal imaging, as well as the limitations associated with the use of standard systems (i.e., firefly luciferase (Luc2) and D-luciferin (D-LH2)) in deeper tissues because of photon absorption. Reviewer #1 also understands that this could be overcome by using systems that produce longer wavelength photons. Because of this basic understanding, they were able to appreciate that one component of our engineered system produces predominantly near IR signals. They could therefore also appreciate that the longer wavelength emission with our systems enabled successful imaging in black fur mice as well as in the deep brain.

The reviewer felt the manuscript needed tighter focus on the major findings. He/she specifically suggested we take some of the emphasis away from the in vitro data because much of it was generated using a different detection device that has different spectral sensitivity compared to a CCD-based animal imaging device. We have reduced the emphasis on differential spectral sensitivity between different devices by mentioning just a single time and providing ample references for those readers who are more interested in these details (page 5, line 128 and page 7, line 210).

Reviewer #1 also specifically suggested that we do more thorough comparison studies in mouse brain tomography (i.e., Figure 5). We have addressed this by repeating experiments and including what we believe to be the most appropriate benchmark, CBR2opt and D-LH2, as suggested also by reviewer 3 (page 9, lines 266-285). Since multispectral bioluminescence tomography feasibility and accuracy mostly depends on the amount of light that can be revealed in the different filters we are now showing that our new bioluminescent system (CBR2/NH2-NpLH2) can be used for this purpose and that the reconstruction of light source is as accurate as using common D-LH2 (new figure 5). However, our system has a clear advantage over D-LH2 due to the fact that it is able to distinguish relatively close light sources (Figure 6). This gain on spatial resolution is because tissues scatter less light at longer wavelengths.

Other suggestions from Reviewer #1 were addressed as follows:

1. They requested we include the DNA sequence of CBR2opt compared to CBR2. We have added appropriate text in the manuscript (page 5, line 157) and also included an alignment between the two sequences in the Supplementary Information. We have also indicated in the text that the

codons in CBR2opt were modified according to their natural frequency found in mice. Finally, we report that the CBR2opt is 79% identical to CBR2.

2. They suggested we be more quantitative in how we describe data in the Results section. We have removed all vague comments and replaced accordingly with more quantitative wording.
3. They suggested we consider using an abbreviation for the novel substrates that would indicate that they contained a naphtho[2,1]thiazole ring system. We believe that our abbreviations (OH-NpLH2 and NH₂-NpLH2) are sufficiently descriptive because they specify hydroxy- or amino-substituted naphthyl rings in association with LH2. We appreciate this reviewer suggesting we use the term naphtho[2,1]thiazole to describe the extra ring system, and have incorporated the term into our description of the novel substrates with regards to their original design (page4, line 109).
4. This reviewer requested evidence supporting our claim of high enantiopurity regarding the synthesis of the novel substrates. We provided HPLC evidence in the supplementary information to confirm the high enantiopurity.
5. The reviewer suggested we replaced the word “theory” with “hypothesis”. We have done so (page 5, line 152).
6. The reviewer suggested we be clearer with our definition of DLSA. In the Methods section, we now define DLSA more appropriately as 5'-O-[N-(dehydroLuciferyl)-sulfamoyl] adenosine (page 12, lines 350-360). In addition, we now refer to it as an active site-bound analog rather than a high-energy intermediate.
7. The reviewer suggested we include NMR spectra for the novel chemistry. We have done so as Supplementary information.
8. The reviewer suggested better clarity in how we communicate the number of replicates in figure legends. We have changed all N=3 to n=3 (convention). We also used this request as an opportunity to be clearer with our statistical description of data. It is now much more obvious throughout the manuscript that our data is presented as means +/- standard deviation.

Reviewer #2 response:

Reviewer #2 also understood the significance of the problem our novel luminescence system addresses, and feels that the large body of work presented was carefully done and that the data we present fully supports our claims of improved performance in animals. The major concern from this reviewer was that our gains in performance are too incremental.

Though the gains we have achieved are not enormous in brain, we have an 8-fold higher signal in black fur mice, a very challenging model that no other paper is showing. We respectfully disagree with this reviewer that the new enzyme and substrates do not represent a novel bioluminescence system. The most red-shifted of our systems is >50 nm longer than anything previously reported for use in animals. Furthermore, the system is truly enabling for multispectral bioluminescence tomography with accuracy. We have now also shown when it can be an advantage (as requested by the other reviewers).

Specific suggestions for revision from Reviewer #2 were as follows:

1. The reviewer suggested citing the recent work of Anderson et al (2017) and Wu et al (2017) for their achievements in shifting luminescence to longer wavelengths. We have added Anderson reference in the introduction among examples of near infrared emitting analogues (page 1, line 75). We did not take into consideration the work of Wu et al. because we considered their experimental design too different from ours and impossible to reproduce. First, it is not possible to find in Wu et al. the depth of implantation of cells in brain or the number of cells and it is clear that the reported fold difference in light signal might be due to more cells being implanted and more superficially. In addition, the 20-fold difference they report is limited to the concentration range used. Moreover, their substrate is not a near infrared emitting analog so we decided to compare our system to Aka-HCl instead (i.e., a commercially available substrate).
2. The reviewer suggested we compare our tomography results to other non-invasive methods of brain imaging for cancer. Following progression of brain tumor xenografts does not represent a challenge anymore in the molecular imaging field. Brain tumor cell lines (e.g. Gli261) can highly express luciferase reporter genes and grow rapidly (3–4 weeks), reaching the humane end-point for the animal (1). BLI is the preferred modality for preclinical studies thanks to high sensitivity, low cost, and medium throughput (2,3). Although giving higher resolution, MRI overly increases the cost and time of the experiments while PET imaging using reporter genes has comparable sensitivity to bioluminescence imaging but it requires more expensive settings. The different modalities are nowadays considered complementary. On the other hand, our model represents the challenge of visualizing transplanted cells in brain like stem cells that are known to decrease in number after transplantation and to migrate in different location (4). This is a field that also has high relevance for Nature readership. We have shown that we can not only efficiently visualize 1×10^5 HEK cells at 3 mm depth in brain, but we were also able to follow their migration with better spatial resolution.

3. Line 76. The reviewer suggested we quantify and qualify our statement that weak signals for near IR systems is limiting. We have done so since we cite 4-5 papers where near infrared emitting analogs are presented and show increased relative sensitivity at specific concentration ranges (page 1, lines 74-78). As mentioned, bioluminescence imaging is mostly chosen by researchers for its high sensitivity. Sensitivity depends on the amount of photons generated in the so-called optical imaging windows (above 650nm) where light can better penetrate tissues. We partially agree with the reviewer that brightness is not the only factor that matters. A bioluminescent system that shows improvement in sensitivity of imaging compared to the most commonly employed Luc2/D-LH2 couple is desirable and will have an impact on the field, independently if this is achieved via spectral shifting or incremental gains in brightness (which results in more photons emitted in the optical imaging windows). Literature demonstrates that luciferase mutants showing spectral shifting associated with higher brightness increase the sensitivity of imaging (5,6,7). On the other hand we also recognized that new bioluminescent systems with different spectral emission will be extremely useful for multiplexing and refining animal experiments or achieving improved resolution as shown in our work and described in the discussion section (page 10-11; lines 294-326).
4. Line 80. The reviewer suggested we modify a sentence about improved signals from Aka-HCl being observed previously only at low concentration. Our intention was to differentiate this type of advantage from one where substrates that are compared are used at concentrations much lower than their most favorable conditions (i.e., highest possible concentration based on solubility). We make the point elsewhere in the paper (page 11, lines 315-316), and decided it would be most appropriate to remove the comment in question.
5. Lines 107-10. The reviewer requested a reference on the design of our analogs including the potential for better photo-stability. We have added an appropriate reference here (Gorka et al (2014)).
6. Lines 112-13. The reviewer requested more detail on chemical synthesis and enantiopurity. We have added a reference to our synthesis schemes 1 and 2 in the Methods section. We have also included data on enantiopurity in the Supplementary Information.
7. Lines 113-14. The reviewer asked about salt forms for the substrates being tested. They were all monopotassium salts and we have made this more apparent throughout the manuscript. The synthesis of all compounds is now described in significant detail in the Methods section.
8. Line 125; The reviewer pointed out that the emission peak for our most red-shifted substrate was 55 nm longer than anything previously reported, rather than the 60 nm difference we described. We

thank the reviewer for pointing this out. In fact, it should be 54 nm, so this is what we are now reporting (now page 5, line 124).

9. Line 128-29. The reviewer makes essentially the same suggestion as Reviewer #1 does with regard to our commentary on PMT bias and luminometer versus CCD-based imaging instrumentation. We believe our response to Reviewer #1 addresses these concerns.
10. Line 144-45. The reviewer questions why we modeled the OH- analog. We have added a comment to the effect that it was chosen because it is the most red-shifted of the two novel substrates. The reviewer also questioned our comments on significance of H-bonds to amino luciferins. We were actually not referring specifically to the H-bonds to amino luciferins, but rather referring to the previous work done on amino luciferins that justified our approach to target residues in luciferases that are involved in H-bonding networks. To make this clearer we have added a sentence at the end of this paragraph in question (page 5, line145).
11. Line 161-62. Same as point 9 above. We have addressed this in the revised manuscript.
12. Line 291
 - a. At this point in the referee's review they deviate to a few different unrelated points. The first has to do with our computational and biological techniques, where they deferred to the other referees to evaluate. We assume that between the three reviewers their expertise in computational and biological methods is adequate for reviewing our work. Based on the comments we believe this to be true.
 - b. The reviewer has concerns that it would not be possible to repeat our chemical synthesis methods by a practicing organic chemist. Though the comment is vague, we have attempted to add more detail to our chemical synthesis descriptions in the Methods section. It is noteworthy that in contrast to this, Reviewer #3 commented that the "manuscript is written clearly with methods that others in the field could reproduce."
13. Line 570; The reviewer suggested S2 and S3 be changed to the correct orientation. We have done so in the revised manuscript.
14. Lines 571 and 574; The reviewer suggested the carboxyl stereocenter should be drawn with stereochemistry. We have done so in the revised manuscript.
15. The reviewer requested more detail for synthetic product yields, optical purity, volumes, and structural characterization by NMR and MS. We now provide yield data and are including HPLC-based enantiopurity analysis for optical purity. We have also added more data on volumes and

include ¹H NMR spectra to support identity and purity. We believe this characterization is satisfactory for the chemical intermediates.

Additional comments from Reviewer 2 were as follows:

16. NMR J values; We now provide NMR J values for multiplicities.
17. The mass of S2 obtained, its percentage yield and melting point need to be stated. S3 the isolated form needs to be stated and if a solid the melting point also. S4 percentage yield and melting point should be stated. Line 621 isolated form needs to be stated, possible melting point and full characterization data. S6percentage yield and melting point needs to be stated. S7 form of final product, possible melting point and percentage yield missing. S8 what form was isolated, possible melting point, mass isolated and percentage yield all missing.
18. Line 653 7-methoxy (?)
19. Line 655.How was neutralization measured. What form was isolated, possible melting point, mass isolated and percentage yield all missing. The optical rotation needs to be stated. As does the optical purity, how it was determined and that evidence submitted as supplementary material.

Points 16–19 above have all been addressed in the experimental section of the revised manuscript. However, since the isolated compounds are not in crystalline form, we decided to not provide melting points.

20. Reviewer 2 requested optical rotation be stated and how it was determined to be explained. The HPLC-based enantiopurity analysis can address the optical purity requirement. The HPLC traces are provided in Supplementary information. We feel the addition of optical rotation would not add any value to the integrity of the publication.

Reviewer #3 response:

Reviewer #3 was generally positive, and like Reviewer #1 clearly understood the important need for longer wavelength emitting (i.e., near IR) luminescence systems for animal imaging.

Specific suggestions for revision from Reviewer #3 were as follows:

1. They suggested more statistical analysis with data presented in graphical form to support statements made in the text. We have done so in plots relating to live cell experiments (Figure 2) and animal imaging data (Figs. 3, 4, and 5)
2. The reviewer suggested we clarify our choices for route of administration when treating animals with substrates. When comparing sensitivity of detection it is important to choose the route that ensures the higher influx of substrate (in case of brain imaging is intravenous injection of substrate). When performing tomography it is important to have a temporal window where in vivo signals are stable in order to collect sufficient light using the different filters (in case of brain imaging is intraperitoneal injection of substrate). We have modified the manuscript accordingly (page 8, line 250 and page 9, line 269) and added information in material and methods (page 18; line 569).
3. We show our data of bioluminescence generated by cells injected in the brain using photon flux since we injected the same amount of cells in every animal for both Luc2 and CBR2opt expressing cell line. We have generated stable cell lines that have the same average expression of GFP. We count the cells and prepare the solution for injection and check for GFP signal before starting experiment (that is our validation) since we seriously doubt that any methods to count cells after injection would be more precise. However, we acknowledge that this information should be stated in the materials and method section and we have now done so (page 18, line 551). The cells injected in black mice were imaged same day and the one injected in brain the day after. We did not perform any tumor study. As stated in the material and methods section a 2-fold signal difference is a 50% percent difference in signal output that allows us to use just 3 mice per group and achieve statistically significance.
4. We agree with the reviewer that for the experiments shown in Figure 5 an additional comparison was needed. We also found the suggestion of using CBR2opt/D-LH2 couple to compare to CBR2opt/NH2-NpLH2 very appropriate. Challenged by this concern, we have performed a new experiment where we compare the accuracy of reconstruction of single light sources in brain using CBR2opt with either D-LH2 or NH2-NpLH2 (New Figure 5). To make calculation of depth more accurate we co-registered our BLI reconstructed images with CT images and calculated the depth of implantation. Moreover, we repeated acquisitions after 5 days when cells should have migrated from their original location. These experiments allowed us to clearly identify the advantage of CBR2opt/NH2-NpLH2 in the reconstruction of two adjacent light sources (New Figure 6). We were able to validate these results by ex vivo analysis and now report the location of the cells in the brain.

REFERENCES

1. Bioluminescence Imaging of an Immunocompetent Animal Model for Glioblastoma. Clark AJ, Fakurnejad S, Ma Q, Hashizume R. *J Vis Exp*. 2016 Jan 15;(107):e53287.
2. In Vivo Molecular Bioluminescence Imaging: New Tools and Applications. Mezzanotte L, van 't Root M, Karatas H, Goun EA, Löwik CWGM. *Trends Biotechnol*. 2017 Jul;35(7):640-652
3. Bioluminescence imaging: progress and applications. Badr CE, Tannous BA. *Trends Biotechnol*. 2011 Dec;29(12):624-33
4. Assessment of therapeutic efficacy and fate of engineered human mesenchymal stem cells for cancer therapy. Sasportas LS, Kasmieh R, Wakimoto H, Hingtgen S, van de Water JA, Mohapatra G, Figueiredo JL, Martuza RL, Weissleder R, Shah K. *Proc Natl Acad Sci U S A*. 2009 Mar 24;106(12):4822-7
5. Red-shifted *Renilla reniformis* luciferase variants for imaging in living subjects. Loening AM, Wu AM, Gambhir SS. *Nat Methods*. 2007 Aug;4(8):641-3
6. A bright cyan-excitable orange fluorescent protein facilitates dual-emission microscopy and enhances bioluminescence imaging in vivo. Chu J, Oh Y, Sens A, Ataie N, Dana H, Macklin JJ, Laviv T, Welf ES, Dean KM, Zhang F, Kim BB, Tang CT, Hu M, Baird MA, Davidson MW, Kay MA, Fiolka R, Yasuda R, Kim DS, Ng HL, Lin MZ. *Nat Biotechnol*. 2016 Jul;34(7):760-7.
7. Visualizing fewer than 10 mouse T cells with an enhanced firefly luciferase in immunocompetent mouse models of cancer. Rabinovich BA, Ye Y, Etto T, Chen JQ, Levitsky HI, Overwijk WW, Cooper LJ, Gelovani J, Hwu P. *Proc Natl Acad Sci U S A*. 2008 Sep 23;105(38):14342-6.

Reviewers' comments:

Reviewer #1 states that (s)he is satisfied with the revision in the Remarks to Editor section.

Reviewer #2 (Remarks to the Author):

The authors have responded to my and the other referee's comments. The paper is now more accurate and more easily understood. Unfortunately there are still one major issues I have with the paper, which I think need to be addressed before final acceptance. In addition there are 2 other disagreements I have with the authors rebuttal and also some minor errors or misunderstandings that I think need attention too.

1. The instructions to authors on the Nature website has clear guidelines for the characterisation of chemical materials. In particular:

'Chemical identity for organic and organometallic compounds should be established through spectroscopic analysis. Standard peak listings (see formatting guidelines below) for ^1H NMR and proton-decoupled ^{13}C NMR should be provided for all new compounds. Other NMR data should be reported (^{31}P NMR, ^{19}F NMR, etc.) when appropriate. For new materials, authors should also provide mass spectral data to support molecular weight identity. High-resolution mass spectral (HRMS) data are preferred.'

While the authors have gone some way to writing a more accurate experimental methods and reporting of the ^1H NMR spectral data, the only new materials to have ^{13}C NMR and MS analysis are the two final compounds OH-NpLH2 and NH2-NpLH2. I think the manuscript is lacking in the adequate characterization data for all the new compounds synthesized in terms of ^{13}C and MS data. This would be required by other science and chemistry journals. In addition, if the compounds are a solid they should have a melting point ('Melting-point ranges should be provided for crystalline materials'). Even most powders under the microscope are crystalline. In addition:

Volumes should be mL

Pg 21, line 611, volume of acetone used?

For the intermediate of S2 the ^1H NMR signal @ δ 7.84-7.91 the signal is described as a multiplet where as the NMR shows a singlet.

S3, the experimental does not say what form was isolated.

S4, line 635, what volume of DCM? Line 637, what volume of water?

In S4 the ^1H NMR signal @ δ 8.12-8.03 is reported as a multiplet, yet in the ^1H NMR it is a doublet.

Synthesis of NH2-NpLH2. How was the acid product isolated by HPLC? There are no work up conditions stated. Unless the reaction was applied directly to the HPLC column, which should be indicated if that was the case? The NMR data for this compound is not listed, but the data for the

potassium salt is. There needs to be some clarity in the presentation of what structure in the paper has been characterized. Either provide full characterization data for the free acid or modify schemes to reflect that only the potassium salt was characterized. Some indication as to the yield of the free acid from the preparation should be included. Line 653, what volume of acetonitrile? The ^1H NMR signal @ δ 3.89-3.67 and 3.65-3.48 are reported as a multiplets when from the spectrum they are triplets or double doublets.

In S6 the ^1H NMR signal @ δ 7.22-7.10 is reported as a multiplet when the spectrum shows it is a doublet or double doublet.

S8 the reported ^1H NMR data does not match the spectrum provided. In addition there is evidence of the starting material (OMe $\sim\delta$ 4.00) and spurious peaks δ 2.5-3.5. This compound is certainly not pure.

OH-NpLH2, work up conditions are not stated (as for NH2-NpLH2).

2. The authors claim throughout the manuscript that their analogues produce the most red shifted bioluminescence to date. I did provide the reference to Anderson's recent paper which the authors have referenced (14). In that paper it describes a nr IR luciferin analogue that has a bioluminescence emission of 805 nm. It is weak, but in light of the comments the authors make about sensitivity at those long wavelengths, it is nevertheless real and the claims made in the paper with respect to their compounds being the most red shifted reported are untrue. This statement appears in the abstract, pg 5 (the peak at 760 nm was in cells not in vivo?) and most clearly in the first paragraph of the Discussion section. The authors compounds are amongst the most or, the most measured under particular conditions, which should be stated in the same sentence. I think it would be more genuine to claim the former.

3. I really think the Wu et al (2017) paper deserves some comment. Although the work does not report a nr IR analogue, the paper reports a brighter luciferin that enables better brain imaging. The brighter the luciferin like emission, the more flux in the nr IR and this can be beneficial for some imaging purposes. A qualified comment in this paper would be appropriate, along the lines of what was written for the author's rebuttal.

Other points:

Abstract, line 42 'resolution' misspelt.

Pg4, line 104, should include ref 14. The authors have described a little more how they designed their substrates, but my original criticism was more aimed at why the seemingly 'bent'-naphthalene as opposed to alternative substitution which would give rise to a more logical linear molecule (See structure attached). Was there any particular reason why the linear compounds were not considered/prepared or is the synthesis of the 'bent' compounds easier etc. I think some details here would be informative for any molecular scientist.

Line 110, Aka-HCL does not contain a benzothiazole. iLH2 would match the description here. The high enantiopurity should be quantified. While the data for NH2-NpLH2 has a high 97% ee, the data provided for OH-NpLH2 is not high at 84% ee. The data in the supplementary files is unclear. Is the HPLC of the OH-NpLH2 the sample from the racemization experiment above it.

If that is the case the analysis of the pure compound needs to be included. Please state how the dehydro compound was assigned (presumably LC MS?). Line 115, please quantify 'extended periods of time'. Data should be provided unless it is in the patent, in which case that should be made clear.

Pg 5, 1st sentence. Can we just check the maths. If your signal is 758 nm and ours was reported as 706, the difference is 52 nm. I originally suggested ~55 nm if you wanted to be vague, but I think you should be precise if you want to point this out. The reference to in vivo is misleading as the data refers to in vitro. Line 148, could the residues be labelled in Fig 1b?

Pg 6, line 179, from Fig 1d surely it is 300-fold lower signal? I think you are a factor of 10 out, please check.

Pg 7. Line 214, is the frequency of photons longer than 640 nm higher for Luc2/D-LH2? It really doesn't look it from Fig 2d.

Pg 9, line 280, I think your pictures 6a/6b are the wrong way round.

Supplementary File 2. Is back-thinned the same as back-illuminated? What is S10747-0909?

END

Reviewer #3 (Remarks to the Author):

The revised manuscript appropriately addresses the review of the first submission.

Author response to Reviewer #2 (reviewer comments in italics):

- *1. The instructions to authors on the Nature website has clear guidelines for the characterization of chemical materials. In particular: ‘Chemical identity for organic and organometallic compounds should be established through spectroscopic analysis. Standard peak listings (see formatting guidelines below) for ¹H NMR and proton-decoupled ¹³C NMR should be provided for all new compounds. Other NMR data should be reported (³¹P NMR, ¹⁹F NMR, etc.) when appropriate. For new materials, authors should also provide mass spectral data to support molecular weight identity. High-resolution mass spectral (HRMS) data are preferred. While the authors have gone some way to writing a more accurate experimental methods and reporting of the ¹H NMR spectral data, the only new materials to have ¹³C NMR and MS analysis are the two final compounds OH-NpLH2 and NH2-NpLH2. I think the manuscript is lacking in the adequate characterization data for all the new compounds synthesized in terms of ¹³C and MS data. This would be required by other science and chemistry journals. In addition, if the compounds are a solid they should have a melting point (‘Melting-point ranges should be provided for crystalline materials’). Even most powders under the microscope are crystalline.*

Response: ¹H, ¹³C, and MS analysis for all new compounds is now included in our revised manuscript. See highlighted sections. In addition, we have included melting point analysis for the two new final product luciferins that are the focus of the work. We have not measured melting points for new compounds that were not obtained as ‘crystalline materials’ (even under the microscope). As they are not crystals, melting point determinations for these compounds would be misleading rather than informative.

- *In addition: Volumes should be mL*
Response: We thank the Reviewer for pointing this out and have addressed throughout in our revised manuscript. For consistency we have also changed all ul to uL.
- *Pg 21, line 611, volume of acetone used?*
Response: 20 mL; We have addressed in our revised manuscript; see line 618 (highlighted).
- *For the intermediate of S2 the ¹H NMR signal @ δ 7.84-7.91 the signal is described as a multiplet where as the NMR shows a singlet. S3, the experimental does not say what form was isolated. S4, line 635, what volume of DCM? Line 637, what volume of water?*
Response: We have addressed all in our revised manuscript. DCM and water both 20 mL; now on lines 647 and 649, respectively. See highlights.
- *In S4 the ¹H NMR signal @ δ 8.12-8.03 is reported as a multiplet, yet in the ¹H NMR it is a doublet. Synthesis of NH2-NpLH2. How was the acid product isolated by HPLC? There are no work up conditions stated. Unless the reaction was applied directly to the HPLC column, which should be indicated if that was the case? The NMR data for this compound is not listed, but the data for the potassium salt is. There needs to be some clarity in the presentation of what structure in the paper has been characterized. Either provide full characterization data for the free acid or modify schemes to reflect that only the potassium salt was characterized. Some indication as to the yield of the free acid from the preparation should be included.*
Response: We initially isolated NH₂-NpLH₂ in free acid form by preparative HPLC without any workup procedures. The updated experimental section now reflects this. The isolated product in free acid form tends to be contaminated with some impurities and the conversion to the K salt helped to clean it up. We report the combined yields and the updated experimental section reflects this. In addition, the chemical synthesis scheme for both final analogs has been updated accordingly.

- *Line 653, what volume of acetonitrile?*
Response: 4 mL. We have addressed this in our revised manuscript; see line 667 (highlighted).
- *The ¹H NMR signal @ δ 3.89-3.67 and 3.65-3.48 are reported as a multiplets when from the spectrum they are triplets or double doublets.*
Response: We have addressed this in our revised manuscript (highlighted).
- *In S6 the ¹H NMR signal @ δ 7.22-7.10 is reported as a multiplet when the spectrum shows it is a doublet or double doublet. S8 the reported ¹H NMR data does not match the spectrum provided. In addition there is evidence of the starting material (OMe ~d 4.00) and spurious peaks δ 2.5-3.5. This compound is certainly not pure. OH-NpLH2, work up conditions are not stated (as for NH₂-NpLH2).*
Response: The spectra data file has been updated to correct the inconsistency. The ¹H NMR of S8 in CD₂Cl₂ is provided with no evidence of major impurities. The description has been updated to reflect the K salt formation. With regard to the description of chemical shifts in general, although we are confident that our NMR spectra support our claims on compound identities, in certain situations, the resolution was not good enough to provide accurate coupling constants due to limitations of the instrument. In these cases, we feel providing a range of chemical shifts is appropriate and is common practice.
- *2. The authors claim throughout the manuscript that their analogues produce the most red shifted bioluminescence to date. I did provide the reference to Anderson's recent paper which the authors have referenced (14). In that paper it describes a nr IR luciferin analogue that has a bioluminescence emission of 805 nm. It is weak, but in light of the comments the authors make about sensitivity at those long wavelengths, it is nevertheless real and the claims made in the paper with respect to their compounds being the most red shifted reported are untrue. This statement appears in the abstract, pg 5 (the peak at 760 nm was in cells not in vivo?) and most clearly in the first paragraph of the Discussion section. The authors compounds are amongst the most or, the most measured under particular conditions, which should be stated in the same sentence. I think it would be more genuine to claim the former.*
Response: We agree with Reviewer #2 that the language surrounding our claims of red-shift significance is too strong and have modified accordingly in each of the specific parts of the manuscript pointed out by this reviewer (see highlights in revised manuscript). We now refer to our new system in the abstract as “significantly red-shifted,” and at the beginning of the discussion section we have replaced the sentence, “*This system produces the most red-shifted bioluminescence to date, with peak emission at 730 nm for NH₂-NpLH2 and 743 nm for OH-NpLH2,*” with “*The amino compound (NH₂-NpLH2; peak emission at 730 nm) is of particular interest because of its demonstrated utility in mice.*” To further soften the tone of our claims, and at the suggestion of the editor, we have also removed all claims (including in the title) of our work being “novel” and avoided using the words “most,” “best,” and “furthest” throughout the manuscript.
- *3. I really think the Wu et al (2017) paper deserves some comment. Although the work does not report a nr IR analogue, the paper reports a brighter luciferin that enables better brain imaging. The brighter the luciferin like emission, the more flux in the nr IR and this can be beneficial for some imaging purposes. A qualified comment in this paper would be appropriate, along the lines of what was written for the author's rebuttal.*
Response: We remain concerned that the in vivo experimental protocols of Wu et al. are somewhat flawed. However, we agree it represents another substrate that might have benefits for

some imaging purposes. Therefore, we have now cited this reference in the discussion section of our revised manuscript (see revised manuscript; highlighted; lines 322-4).

Other points:

- *Abstract, line 42 'resolution' misspelt.*
Response: We thank the Reviewer for finding this typographical error and have corrected it in revised manuscript.
- *Pg4, line 104, should include ref 14. The authors have described a little more how they designed their substrates, but my original criticism was more aimed at why the seemingly 'bent'-naphthalene as opposed to alternative substitution which would give rise to a more logical linear molecule (See structure attached). Was there any particular reason why the linear compounds were not considered/prepared or is the synthesis of the 'bent' compounds easier etc. I think some details here would be informative for any molecular scientist.*
Response: We appreciate Reviewer #2 being curious about the design of our new substrates, and whether we considered closely related analogous compounds (e.g. linear forms). In fact, we did consider the 'linear' analogs of these compounds and actually synthesized the linear version of OH-NpLH2. This strategy was ultimately not pursued because this analog had lower solubility than the bent forms and was less active as a luciferase substrate. It is our opinion that this extra discussion will undoubtedly serve to address Reviewer #2's curiosity, but that it is beyond the scope of what is needed on this topic for a broad readership. Furthermore, we are concerned that going into these details in the manuscript will dilute our primary messages regarding the importance of extending the Π -conjugation systems to achieve longer wavelength emission and the performance of these compounds in relevant in vivo models.
- *Line 110, Aka-HCL does not contain a benzothiazole. iLH2 would match the description here.*
Response: We thank the Reviewer for pointing this out and have corrected this in our revised manuscript (highlighted on line 110). We have correctly replaced "benzothiazole" with "aryl."
- *The high enantiopurity should be quantified. While the data for NH2-NpLH2 has a high 97% ee, the data provided for OH-NpLH2 is not high at 84% ee. The data in the supplementary files is unclear. Is the HPLC of the OH-NpLH2 the sample from the racemization experiment above it. If that is the case the analysis of the pure compound needs to be included. Please state how the dehydro compound was assigned (presumably LC MS?).*
Response: We have included both the calculated enantiopurity and the data used to determine it for both compounds in our revised manuscript. To address Reviewer #2's questions about protocols more specifically, we conducted forced racemization experiments using a sample with high enantiopurity to enrich the L enantiomer for establishment of chiral resolution. During the heat treatment, the dehydro compound, confirmed by LC-MS and UV abs, was observed as well. The HPLC analysis clearly confirmed the observation. Subsequently, the D-OH-NpLH2 sample with high enantiopurity was analyzed under the identical analytical condition and the HPLC trace was provided. The sample has been stored in -20 °C freezer for more than 4 years, and frequently underwent thaw-freeze processes.
- *Line 115, please quantify 'extended periods of time'. Data should be provided unless it is in the patent, in which case that should be made clear.*
Response: We have modified to say "at least 24 h"; see revised manuscript (highlighted; line 115).

- Pg 5, 1st sentence. Can we just check the maths. If your signal is 758 nm and ours was reported as 706, the difference is 52 nm. I originally suggested ~55 nm if you wanted to be vague, but I think you should be precise if you want to point this out. The reference to in vivo is misleading as the data refers to in vitro.*

Response: We thank the Reviewer for pointing this out. We agree it was a source of confusion and have changed to precise values and removed the reference to in vivo; see revised manuscript (highlighted; lines 123-5).
- Line 148, could the residues be labelled in Fig 1b?*

Response: We agree with the Reviewer that it would be better to have the residues labeled. They are labeled in the new version of Figure 1b.
- Pg 6, line 179, from Fig 1d surely it is 300-fold lowers signal? I think you are a factor of 10 out, please check.*

Response: We thank the Reviewer for commenting on this. It is actually 33-fold and we have modified the text in the revised manuscript accordingly so that it is clearer (highlighted; lines 178-81).
- Pg 7. Line 214, is the frequency of photons longer than 640 nm higher for Luc2/D-LH2? It really doesn't look it from Fig 2d.*

Response: We thank the Reviewer for pointing this out and have re-written this sentence to make it clearer for the reader (highlighted; lines 214-17).
- Pg 9, line 280, I think your pictures 6a/6b are the wrong way round.*

Response: We thank the reviewer for noticing this. Instead of modifying the figure we have changed the sentence in the manuscript (see highlighted manuscript; lines 282-84). The figures now match the text.
- Supplementary File 2. Is back-thinned the same as back-illuminated?*

Response: Indeed, back-illuminated cameras are thinned. In the back illuminated CCD camera design light falls onto the back of the CCD in a region where the bulk of the silicon has been thinned by etching until it is transparent (a thickness corresponding to about 10–15 microns). To make this clearer we have revised Supplementary File 2.
- What is S10747-0909?*

Response: The S10747-0909 is a back-illuminated CCD camera sensor that delivers drastically improved near-infrared sensitivity by the widened depletion layer. However, this model is not cooled so not recommended for in vivo imaging where long exposures are required. We have added additional information to this point in Supplementary File 2.

Reviewers' Comments:

Reviewer #1 (Remarks to the Author):

I offer these comments regarding the responses to Reviewer #2. The authors have now included a sufficient amount of NMR and MS data in the revised experimental and in File #5 – NMR Spectra. Also, new spectral data are included, presumably from a different sample, on a compound that previously contained impurity peaks. Only a few very minor issues remain. It isn't clear why the authors are reluctant to carry out simple melting point analyses on all samples as requested. Melting points of organic solids are useful even if the samples are not crystalline as long as the history of the purification (chromatography, recrystallization solvent, sublimation, etc.) is described. The possible exceptions are compounds that decompose (this may be the case with the K⁺ salts), but it is still normal practice to include melting point data. Also, on line 609 of the Materials and Methods "Varian" is spelled incorrectly. It is my opinion that the revised manuscript is acceptable for publication.

Author response to Reviewer #1 (reviewer comments in italics):

I offer these comments regarding the responses to Reviewer #2. The authors have now included a sufficient amount of NMR and MS data in the revised experimental and in File #5 – NMR Spectra. Also, new spectral data are included, presumably from a different sample, on a compound that previously contained impurity peaks. Only a few very minor issues remain. It isn't clear why the authors are reluctant to carry out simple melting point analyses on all samples as requested. Melting points of organic solids are useful even if the samples are not crystalline as long as the history of the purification (chromatography, recrystallization solvent, sublimation, etc.) is described. The possible exceptions are compounds that decompose (this may be the case with the K⁺ salts), but it is still normal practice to include melting point data. Also, on line 609 of the Materials and Methods "Varian" is spelled incorrectly. It is my opinion that the revised manuscript is acceptable for publication.

Response: Though possible, it would require several weeks of effort to make the intermediates and do the melting analysis. Our bigger concern with doing this analysis, however, has to do with the fact that we respectfully disagree with Reviewer 1 that this information will add any value to the manuscript. As we have previously disclosed, all of our novel chemical intermediates are non-crystalline in nature. We have also previously expressed our concern that doing the melting analysis on these non-crystalline materials is likely to be misleading rather than informative.

Melting point analysis has historically been used by chemists to set references for compound identity and for evaluating compound purity (based on a melting point range). But in the modern era, NMR, LC-MS and other more sophisticated methods provide the exact same information about identification and purity, but with significantly more detail and accuracy. Because these newer methods have made melting point analysis far less critical, most chemistry-related journals request melting points only for crystalline materials (as does Nature Communications). As another example, Organic Letters (known for its reputation as having the highest standards for chemical characterization) actually instructs authors to NOT report melting points for non-crystalline materials (“A melting point range should be reported for all crystalline compounds. Melting points of non-crystalline amorphous compounds should not be reported.”).

In summary, we believe making new compounds and measuring their melting points will not provide value to the manuscript. Further, the guidelines of Nature Communications clearly state that melting point analysis is only required for novel compounds that are crystalline in nature and take a central part in the study. As our intermediate compounds are non-crystalline and we provided melting points of the two substrates that DO take a central part in the study, we believe that we have met ALL defined requirements as set forth by Nature Communications in regards to chemical characterization.

Finally, we appreciate Reviewer #1 pointing out the typographical error on p. 609 (Materials and Methods). We have corrected this in the revised version of the manuscript.